# Low-Frequency Broadband Absorbing Coatings Based on MOFs: Design, Fabrication, Microstructure and Properties

**Wei Si [1], Qingwei Liao [1,2,3,\*], Wei Hou [1], Liyin Chen [4,\*], Xiaolu Li [1], Zhiwei Zhang [1], Minna Sun [1], Yujun Song [5,\*] and Lei Qin [1,2,3,\*]**

[1] Key Laboratory of Sensors, Beijing Information Science and Technology University, Beijing 100192, China; siwei_2021@163.com (W.S.); houwei112@163.com (W.H.); lixiaolu@bistu.edu.cn (X.L.); zhiweizhang@bistu.edu.cn (Z.Z.); sunminna331@163.com (M.S.)

[2] Key Laboratory of Modern Measurement & Control Technology, Ministry of Education, Beijing Information Science & Technology University, Beijing 100192, China

[3] Key Laboratory of Photoelectric Testing Technology, Beijing Information Science & Technology University, Beijing 100192, China

[4] Harvard John A. Paulson School of Engineering and Applied Sciences, Harvard University, Cambridge, MA 02138, USA

[5] Center of Modern Physics Technology, School of Mathematics and Physics, University of Science and Technology, Beijing 100083, China

\* Correspondence: liaoqingwei@bistu.edu.cn (Q.L.); liyin_chen@g.harvard.edu (L.C.); songyj@ustb.edu.cn (Y.S.); qinlei@bistu.edu.cn (L.Q.)

**Abstract:** Although most microwave absorbing materials (MAMs) have good absorption ability above 8 GHz, they perform poorly in the low-frequency range (1–8 GHz). Metal–organic frameworks (MOFs) derived carbon-based composites have been highly sought after in electromagnetic materials and functional devices, due to their high specific area, high porosity, high thermal stability, low reflection loss, and adjustable composition. In this review, we first introduce the three loss types of MAMs and argue that composite materials are effective ways to achieve broadband absorption. Secondly, the absorbing properties of traditional materials and MOF materials in the literature are compared, followed by a discussion of the promising strategies for designing MAMs with broadband absorption in low frequencies based on the recent progress. Finally, the main problems, fabrication methods, and applications are discussed for their future prospects.

**Keywords:** microwave absorbing materials; low frequency; broadband; composite





## 1. Introduction

Microwave absorbing materials (MAMs) are widely used in many scientific fields, such as communication engineering, medicine, microelectronics, and so on [1–9]. Coating absorbing materials on the fuselage of radar or communication equipment, antenna, and all interference objects around can make them more sensitive and accurate to detect hostile targets [10,11]; Coating absorbing materials on the four peripheral walls of the opening of the radar parabolic antenna can reduce the interference of the side lobe to the main lobe and increase the action distance of the transmitting antenna while reducing the interference of false target reflection to the receiving antenna. The application of absorbing materials in satellite communication systems will avoid the interference between communication lines and improve the sensitivity of satellite communication machines and ground stations, so as to improve the communication quality. The use of MAMs in high-power radar, communication machines, microwave heating, and other equipment can prevent electromagnetic radiation or leakage and then protect the health of operators.

With the rapid development of low dimensional materials, nano-structured MAMs show excellent microwave absorption abilities. Nanostructured absorbing materials have a high specific surface area and rich interfaces, which can provide a variety of absorbing

mechanisms such as interface effect, high anisotropy ratio, conduction network, magnetic coupling, magnetic dielectric synergistic effect, and so on, which greatly improves the absorbing efficiency. As far as the reported nanostructured materials are concerned, most of them have absorbing properties at 8–12 GHz or 8–18 GHz reflection loss (*RL*) is less than—10 dB and absorption rate is greater than 90%), and relatively few have absorbing properties at low frequencies (1.2−8 GHz) and can achieve broadband absorption (greater than 5 GHz). The key to determining the bandwidth of absorbing frequencies is impedance matching regulation. MOFs derivatives obtained by pyrolysis of MOFs composed of metal atoms and organic ligands in the air or inert atmosphere have a high specific surface area, high porosity, and adjustable chemical structure [12–19]. Since MOF derivatives containing magnetic metal ions are a composite of magnetic nanostructured materials and nanostructured carbon materials after pyrolysis, the proportion of magnetic materials and dielectric materials can be adjusted through the allocation of raw materials. Therefore, MOF materials are a kind of interesting and effective MAMs, and it is also a research hotspot in the field of microwave absorption [20].

This paper mainly summarizes the EM absorption performance of MOFs in low-frequency bands. Firstly, the absorbing mechanism and classification of absorbing materials are introduced, and the advantages of MOFs derivatives and traditional materials in improving low-frequency and broadband absorption are described. The corresponding solutions are proposed for achieving broadband absorption in low frequency. Finally, by summarizing the MAMs with microwave absorbing ability at low frequencies published in recent years, the laws of particle structure and preparation methods are analyzed, hoping to provide new ideas for future design strategies, manufacturing methods, and applications.

## 2. Basic Theories Related to MAM Research

### 2.1. Impedance Matching Characteristic

Figure 1 shows the schematic diagram of the EM wave absorption mechanism. When EM waves are emitted from air to a composite material, some of them will reflect from or transmit through the material, while others will be absorbed by the material. The EM wave entering the absorbing material will be reflected many times, attenuated, and converted into heat energy [21–23].

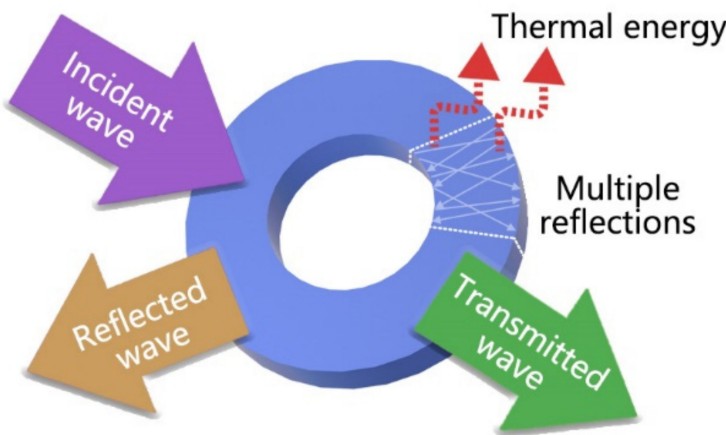

**Figure 1.** Schematic diagram of EM wave absorption mechanism.

Better impedance matching allows more EM waves to enter the material, instead of being reflected from the surface. The reflectivity R of EM waves at the interface of the two media can be expressed as [24]:

$$R = \frac{z_2 - z_1}{z_2 + z_1} \tag{1}$$

Here, $z_1$ is the impedance of free space. $z_1 = \sqrt{\mu_0/\varepsilon_0}$, where $\mu_0$ and $\varepsilon_0$ are the permeability and permittivity in the hollow, respectively; $z_2$ is the intrinsic impedance of the absorbing material. $z_2 = \sqrt{\mu_r/\varepsilon_r}$, where $\mu_r$ and $\varepsilon_r$ are the complex permittivity and the complex permeability of the absorbing material, with expressions of the following [25,26]:

$$\varepsilon_r = \varepsilon' - j\varepsilon'' \tag{2}$$

$$\mu_r = \mu' - j\mu'' \tag{3}$$

where $\varepsilon'$ and $\mu'$ are the real parts of $\varepsilon_r$ and $\mu_r$, representing the energy storage capacities of the electric field and the magnetic field. $\varepsilon''$, $\mu''$ are the imaginary parts, representing the electrical and magnetic field loss capacities of the material.

According to (1), in order to achieve strong EM wave absorption, $|z_1/z_2|$ should be as close to one as possible. This enables the complete entry of incident EM waves, and the dissipation of energy can occur with the premise of entry. The former condition can be evaluated by the general matching rule [27]:

$$\varepsilon'\mu'' - \varepsilon''\mu' = 0 \tag{4}$$

### 2.2. Attenuation Characteristic

The $RL_{min}$ and EAB of a material are important criteria for evaluating the absorption performance of the material. The expression of $RL$ is [28]:

$$RL = 20lg\frac{z_{in} - z_0}{z_{in} + z_0} \tag{5}$$

where $z_{in}$ is the effective input impedance $z_{in} = \sqrt{\frac{\mu_r}{\varepsilon_r}} \tanh(j2\pi\frac{fd}{c}\sqrt{\mu_r})$, $f$ is frequency, $d$ is the thickness of the medium, and $c$ is the velocity of light. Equation (5) can be simplified to $RL = 20\log(P_a)$, where $P_a$ is the reflected power of EM wave on the surface of the material. When $RL$ is less than $-10$ dB, the absorbing capacity of the material can reach 90%, so the frequency range of $RL \leq -10$ dB is defined as the EAB. Of course, the smaller the $RL$ value at a particular frequency, the better the absorbing performance is at that frequency.

Materials dissipate the incoming EM energy by converting it into heat energy or other forms of energy in different ways. The expression of its attenuation coefficient is [29]:

$$\alpha = \frac{\sqrt{2\mu'\varepsilon'}\pi f}{c} \cdot \sqrt{(tan\delta_\varepsilon tan\delta_\mu - 1) + \sqrt{(tan\delta_\varepsilon tan\delta_\mu - 1)^2 + (tan\delta_\varepsilon + tan\delta_\mu)^2}} \tag{6}$$

where $tan\delta_E = \varepsilon''/\varepsilon'$ and $tan\delta_M = \mu''/\mu'$ are dielectric and magnetic loss tangents, respectively [30–32]. Based on the formula of loss tangent, the loss capacity of materials can be improved by increasing the imaginary part of complex dielectric constant and complex permeability or by decreasing the real part [23]. Therefore, exceptional EM wave absorption results from the strong dielectric and magnetic loss capacities as well as the satisfactory balance between the two [27].

In addition, the thickness also has a great influence on the absorption performance. The frequency corresponding to the $RL_{min}$ usually moves to lower frequencies with increases in material thickness. This result could be explained by the interference theory [33]. When the relationship between the absorbing specimen thickness d and the wavelength of the electromagnetic wave meets (7), peaks would occur in the curves.

$$d = (2n+1)\frac{\lambda}{4} = (2n+1)\frac{\lambda_0}{4\sqrt{\mu_r\varepsilon_r}} \quad (n = 0,1,2,3\ldots) \tag{7}$$

where $\lambda$ and $\lambda_0$ are the wavelengths of the electromagnetic wave in the absorbing specimen and free space, respectively, and $n$ is an integer. By transforming (7), $\lambda_0$ can be expressed as:

$$\lambda_0 = \frac{4d\sqrt{\mu_r \varepsilon_r}}{(2n+1)} \tag{8}$$

It can be seen from (8) that the number of peaks could increase with an increase in the thickness d. In contrast, the possibility of the wave penetrating the absorbing specimen becomes smaller with an increase in the thickness d, causing a decrease in the interference offset of the reflection wave between the top and the bottom of the layer [34].

### 2.3. Classification of MAM

According to the loss mechanism, MAMs can be divided into dielectric loss and magnetic loss, as shown in Table 1.

**Table 1.** Characteristics of dielectric loss and magnetic loss MAMs.

| Type | Mechanism (Main) | Materials | Disadvantage |
|---|---|---|---|
| dielectric loss $tan\delta_M < tan\delta_E$ | polarization process of electrons, ions, and interfaces. | barium carbonate electric ceramics, silicon nitride, manganese dioxide, etc. | large matching thickness [35,36]. |
| magnetic loss $tan\delta_M > tan\delta_E$ | domain wall resonance, natural resonance, and eddy current loss. | ferrite, carbonyl iron powder (CIP) magnetic metal powder, etc. [37] | prone to oxidize, high density, low saturation magnetization. |

The dielectric polarization includes electron cloud displacement polarization, ion displacement polarization, electric distance shifting polarization of polar media, defect dipole polarization, etc. (Figure 2a). The main attenuation is from the polarization relaxation process, which is due to the lag of polarization time compared to the phase change of the electric field [38–41]. Dielectric loss materials include barium carbonate electric ceramics, silicon nitride, and manganese dioxide, which are mainly absorbed in the frequency range of 0.1 GHz–1 THz. Although they have broad EAB and low density, their matching thicknesses are large, making them difficult to meet the requirements of "thin" in the development of MAMs.

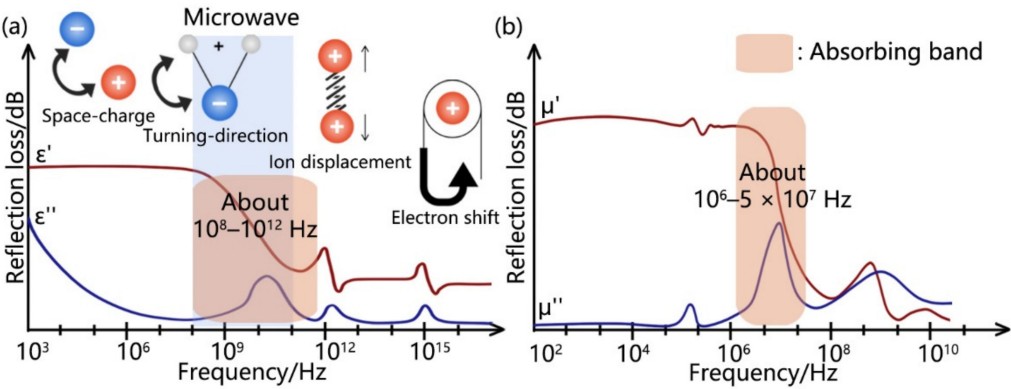

**Figure 2.** (**a**) Complex dielectric constant image and polarization mechanism image of dielectric loss MAMs; (**b**) complex permeability image of magnetic loss MAMs.

The magnetic loss MAM absorbs EM waves through hysteresis loss, domain wall resonance, natural resonance, and eddy current loss in an alternating electric field [35,36]. It can be seen from Figure 2b that the main absorbing frequency of magnetic materials is about 1 MHz–50 MHz. The magnetic loss materials include ferrite, CIP, magnetic metal powder, etc. [37,42,43]. Although magnetic materials have a significant absorbing effect at low-frequency bands, due to the high density of these materials, it is difficult to meet the

requirements of "light" in the development of MAMs. Moreover, the material is limited by the "Snoek limit", the EAB is narrow, and the materials are prone to oxidize, which limits the application of magnetic loss MAMs in many fields.

It is difficult for a single absorber to achieve the balance between dielectric constant and permeability and meet the requirements of impedance matching [44]. A composite material formed by combining two or more materials with different loss types by physical or chemical methods can be: ① adjusted the electromagnetic parameters of the material by changing the material ratio to improve the matching effect of low frequencies; ② maximized the use of various loss mechanisms and combine their corresponding absorption bands to wider absorption bandwidth [45–48]; ③ produced synergistic effect (1 + 1 > 2) and realize the maximum absorption of EM wave energy in a broad frequency range [49–51]; ④ meet the requirements of "light, thin, broad and strong" [52].

## 3. Design Principles of Low-Frequency Broadband MAMs

The study of MAMs broadband absorption in low frequencies mainly aims at two key problems: the design of low-frequency characteristics and the realization of broadband characteristics.

### 3.1. Design of Low-Frequency Characteristics

In general, the MAMs are divided into magnetic loss and electrical loss. Magnetic loss is often much less than electrical loss, so microwave absorbing materials mainly rely on electrical loss to dissipate EM waves. However, EM waves first need to enter into the materials, not to be reflected or scattered, and then to achieve impedance matching with the EM wave. Impedance matching often requires magnetic regulation. Therefore, the absorption frequency range mainly depends on the design of the magnetic part. The absorption frequency of materials is directly dependent on the particle size. Related to the absorption frequency of EM wave, there are the following size problems: the skin depth determines the maximum size of electromagnetic wave energy; the size of a single domain determines the minimum size of electromagnetic wave energy; quantum size effect, resonant absorption of microwave quantum and microwave absorbing unit. As shown in Figure 3, for frequencies below 8 GHz, the skin depth of the absorption unit is 4.5 μm, so the upper limit of the size of the MAMs below 8 GHz is 4.5 μm, which is much larger than that of nanometer materials. In large particles, energetic considerations favor the formation of domain walls. Magnetization reversal thus occurs through the nucleation and motion of these walls. As the particle size decreases toward some critical particle diameter, Dc, the formation of domain walls becomes energetically unfavorable, and the particles are called single domains. Changes in the magnetization can no longer occur through domain wall motion and instead require the coherent rotation of spins, resulting in larger coercivities. As the particle size continues to decrease below the single domain value, the spins are increasingly affected by thermal fluctuations and the system becomes superparamagnetic [53]. For magnetic materials with different crystal structures, the size of a single domain is slightly different, which can be determined by observing the average size of primary crystallinities by high-magnification TEM [15]. Generally, the size range is about 15–25 nm, so the lower limit of the size of MAMs is about 15–25 nm. When the size of the absorption unit is the same as the energy of the microwave quantum, the absorption unit will produce resonant absorption of the incident EM wave. The relationship between the size of the absorption unit and the microwave frequency is shown in Figure 3. The unit size of resonance absorption at the frequency of 1–8 GHz is about 10–25 nm. In addition, reducing the dimension can increase the anisotropy, and the enhancement of anisotropy can accelerate the spin relaxation time to increase the eigenfrequency. At the same absorption frequency, the size limit of low dimensional absorption units can be greatly relaxed to increase the design flexibility. Due to the tip effect and the edge effect, low dimensional magnets usually show strong magnetic coupling interaction. The formation of the magnetic coupling network can significantly improve the magnetic loss ability. To sum up, when

designing low-frequency absorbing materials, low-dimensional (nanofibers or nanoflakes) strong magnetic metal materials above 15 nm can be selected as one of the components of MAMs.

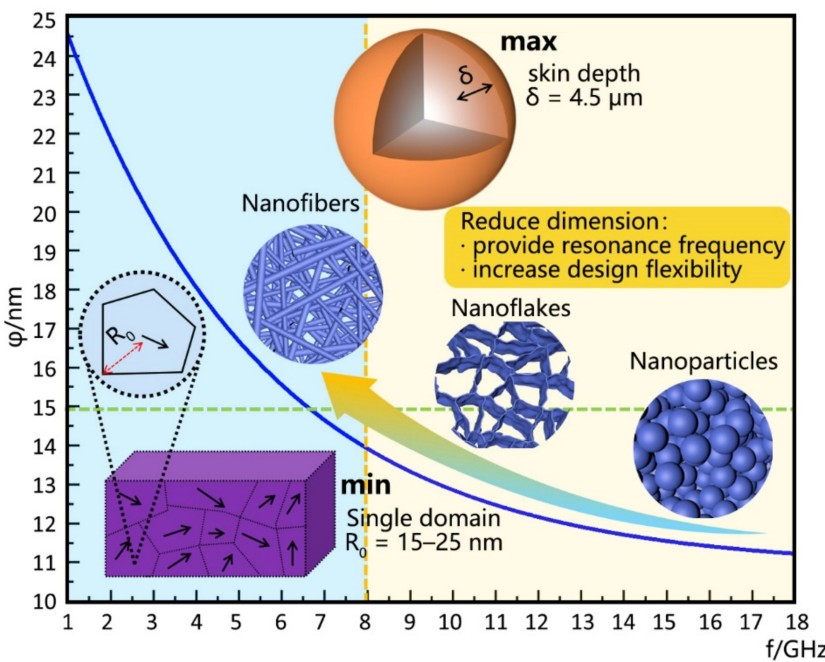

**Figure 3.** Relationship between the size of absorption unit and microwave frequency.

### 3.2. Realization of Broadband Characteristics

From the macroscopic principle, composite is an effective way to obtain materials with broadband absorbing properties. When the materials with different absorption frequencies are combined, the absorption characteristic peaks can be mathematically superimposed, so as to broaden the absorption peaks. The ways of compounding include physical compounding and chemical compounding (Figure 4).

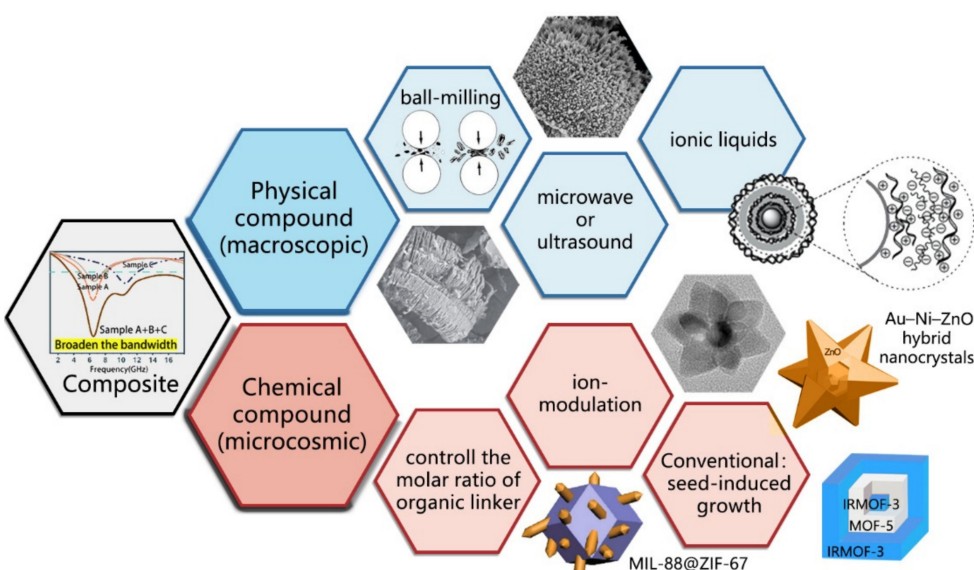

**Figure 4.** Schematic of physical and chemical composition of materials. Image for "ball-milling": Reprinted with permission from Ref. [54]. Copyright 2017, Elsevier. Image for "ultrasound": Reprinted with permission

Physical compounding is the macroscopic mixing of different materials by means of ball milling [54,59], microwave or ultrasound [60], ionic liquids [56], etc. As shown in Figure 5, by mixing MOF derivative materials with good microwave absorption performance in different frequency bands, the effect of improving the EM absorption performance of the overlapping part and broadening the frequency bands are achieved. Table 2 summarizes the absorption properties of the materials shown in Figure 5.

Chemical compounding is the microscopic growth or hybridization of materials or elements with different properties. For microwave absorbing materials, chemical composite is mainly for fine control of microstructure, which is usually designed as a core–shell structure. The core–shell structure absorbing material refers to the multi-level and multi-dimensional composite material designed and prepared on its surface with material as the core. The extensive and close contact between the core–shell and the shell–shell can produce rich interface effects, including interface polarization, scattering effect, and electron band distortion, which can significantly improve the absorbing performance. MOF@MOF can be realized through conventional seed-induced growth procedures, ion modulation strategy [61,62], controlling the mole ratio of each organic linker [63], etc. In addition, the magnetic coupling effect and synergistic effect can be realized by controlling the shell size, which can enhance the microwave absorption effect, and its diversified components also have natural advantages in broadening the absorption bandwidth.

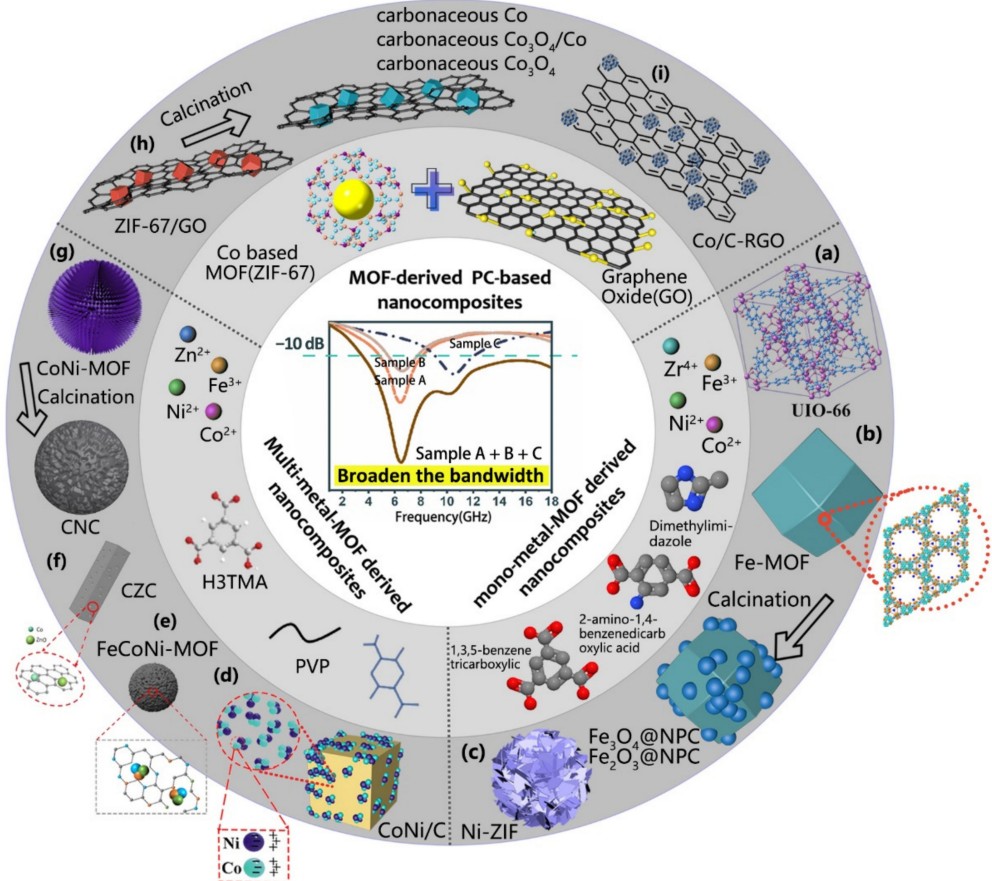

**Figure 5.** The EAB is broadened by the combination of a variety of MOF derivatives. (**a**) Crystal structure of

UIO-66; (**b**) Fe-MOF and its derivatives $Fe_3O_4$@NPC and $Fe_2O_3$@NPC; (**c**) Ni-ZIF. Reprinted with permission from Ref. [64]. Copyright 2019, American Chemical Society. (**d**) CoNi/C; (**e**) FeCoNi-MOF; (**f**) porous Co/ZnO/C(CZC); (**g**) CoNi-MOF and its derivative hollow CoNi@C(CNC). Reprinted with permission from Ref. [65]. Copyright 2019, Elsevier. (**h**) ZIF-67/GO and its derivatives carbonaceous Co, carbonaceous $Co_3O_4$/Co and carbonaceous $Co_3O_4$; (**i**) Co/C-reduced graphene oxide(RGO). Reprinted with permission from Ref. [66]. Copyright 2017, Elsevier.

**Table 2.** Summarizes the absorptive properties of absorbent materials (including samples a to h).

|  | Absorbing Agent | $RL_{min}$ (dB) | Maximum Peak Position (GHz) | Frequency Range (GHz) ($RL \leq -10$ dB) | Efficient Bandwidth (GHz) | Effective Absorption Frequency (<8 GHz) | EAB Range (>5 GHz) | Characteristic | Refs |
|---|---|---|---|---|---|---|---|---|---|
| (a) | $ZrO_2$/C-800 | −58.7 | 16.8 | 11.5–17.0 | 5.5 |  | √ | plentiful, dense, and equally distributed | [67] |
| (b) | $Fe_3O_4$@NPC | −65.5<br>−50 | 9.8<br>6.9 | 7.7–12.2<br>5.4–8.2 | 4.5<br>2.8 |  |  | porous | [68] |
| (c) | Ni@C-ZIF | −86.8 | 13.2 | 6.1–8.1 | 2 |  |  | spherical-like hierarchical 3D nanostructures | [64] |
| (d) | CoNi/C-650 | −74.7 | 15.6 | 10.9–13.2 | 3.3 |  |  | multi-metal | [69] |
| (e) | FeCoNi-MOF-600<br>FeCoNi-MOF-700 | −23.4<br>−69.3 | 6.0<br>5.52 | 5.1–7.9<br>4.7–7.2 | 2.8<br>2.5 | √ |  | multi-metal | [70] |
| (f) | Porous CZC-800 | −21.60 | 5.5 | 4.9–7.0 | 2.1 | √ |  | porous | [71] |
| (g) | CNC-1:1 | −44.8 | 10.7 | 6–7.8 | 1.8 | √ |  | porous and hollow | [65] |
| (h) | $Co_3O_4$/Co/RGO | −52.8 | 13.12 | – | – |  | √ | two-dimensional growth | [72] |
| (i) | Co/C-RGO | −52<br>−27 | 9.6<br>13.48 | 7.2–13<br>10.3–18 | 5.8<br>7.7 |  | √ | two-dimensional growth | [66] |

## 4. Low-Frequency Broadband Characteristics of MOFs

Conductive metal–organic frameworks (MOFs) crystalline extended structures are constructed by stitching together inorganic polynuclear clusters (termed secondary building units (SBUs)) and organic linkers by strong bonds [73–75] (Figure 6). Because the metals in MOFs are covalently linked to the organic ligands, the structures of MOFs are diversified. Common MOFs were shown in Figure 4 [76,77]. During the preparation process, MOF with different morphologies can be obtained by adjusting the metal, organic ligand, and precursor ratio. MOF has the advantages of high porosity, low density, large specific surface area, regular channel, adjustable pore structure, and topological structure diversity [78,79]. Because the metal atoms can be flexibly replaced or a specific solvent in the machining process can be added to change its structure by chemical displacement method. Thus, their structures and composition have a large design degree of freedom. Through the design of structures and composition, the dielectric constant and permeability can be adjusted to realize impedance matching in low frequency.

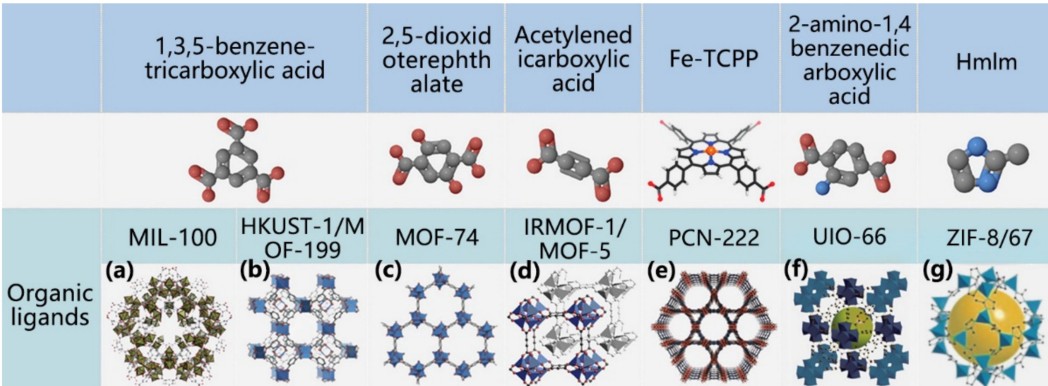

**Figure 6.** Structures of main types of MOFs. (**a**) MIL-100; (**b**) HKUST-1/MOF-199; (**c**) MOF-74. Reprinted with permission from Ref. [80]. Copyright 2011, Elsevier. (**d**) IRMOF-1/MOF-5; (**e**) PCN-222. Reprinted with permission from Ref. [81]. Copyright 2012, John Wiley and Sons. (**f**) UIO-66; (**g**) ZIF-8/67.

### 4.1. Classification of MOF

According to the number of metal elements in MOF materials, they can be divided into single-metal MOFs and multi-metal MOFs.

Cobalt-based MOF precursors have been widely used. The microwave absorbing performance of MAM is highly dependent on the morphology of MOF precursors. As a typical cobalt-based MOF, ZIF-67 has high application potential in the microwave absorbing field due to its ultra-high porosity. However, the morphology of cobalt-based MOF is not limited to the diamond dodecahedron. Liu et al. [82] used polydopamine (PDA) as a carbon source and nitrogen source, and PVP as a protective layer and dispersant to prepare concave cube-shaped nanoparticles by the self-template eco-friendly method (Figure 7a). When the filling ratio of the concave cube sample is 15%, its $RL_{min}$ is −26.8 dB and EAB is 1.4 GHz (4.8–6.2 GHz), and the corresponding thickness is 4.5 mm. As shown in Figure 7b, a novel earthworm-like Co-MOF derived (CO/CoO)@C composite was prepared by facile hydrothermal and annealing route [83]. The relevant microstructures and EM wave absorbing properties could be regulated by the annealing temperature, the optimal performance of the as-prepared composite can be obtained under the annealing temperature of 600 °C, i.e., the $RL_{min}$ is −40.98 dB at 0.89 GHz with relevant efficient absorbing bandwidth of 0.43–1.88 GHz for the thickness of 3.5 mm. Moreover, the widest low-frequency EAB of 0.68–3.0 GHz can be also achieved for only 2.36 mm thickness.

The microwave absorption properties of nickel-based MOF derivatives are derived from the synergistic effect of multiple reflections, magnetic loss, conductive loss, and dipole polarization. Yang et al. [84] synthesized Ni-MOF hollow spheres by hydrothermal method. After carbonization in nitrogen at 600 °C, a surface layer made of different Ni architectures is formed. The shape and size of the surface architectures, (e.g., needles or pillars) can be adjusted effectively by controlling the duration of the hydrothermal reaction (Figure 7c). It is found that the 10 h sample could reach a broad EAB of about 4.8 GHz (4.5–9.3 GHz) and the $RL_{min}$ is −58 dB, with a thickness of 5 mm. In reference [65], Ni-based MOFs derivatives with dimethylimidazole as organic ligands were successfully obtained. The corresponding EAB was 7.4 GHz (4–11.4 GHz) with a thickness range of 1.5 mm to 4.0 mm. The interfacial polarization among Ni and C and N-doping introduced by nitrogen-containing ligands were beneficial to enhance microwave attenuation. The defects produced in the N atom and carbon matrix can be used as polarization centers to introduce more dipole polarization and enhance the EM wave absorption properties.

Zr-based MOF derivatives have excellent chemical stability and environmental friendliness. Due to their low conductivity, they can neutralize the excessive conductivity loss of graphitized carbon, so they play a key role in achieving good impedance matching. In reference [68,85,86], MOF-derived $ZrO_2$/C octahedra have been successfully synthesized from UIO-66 (Figure 7d). This is a benefit of the suitable carbonization treatment, which

leads to a strong attenuation capacity and impedance matching characteristic. Additionally, the EAB could cover 91.3% (3.4–18.0 GHz) of the measured frequency within the thickness range of 1.0–5.0 mm.

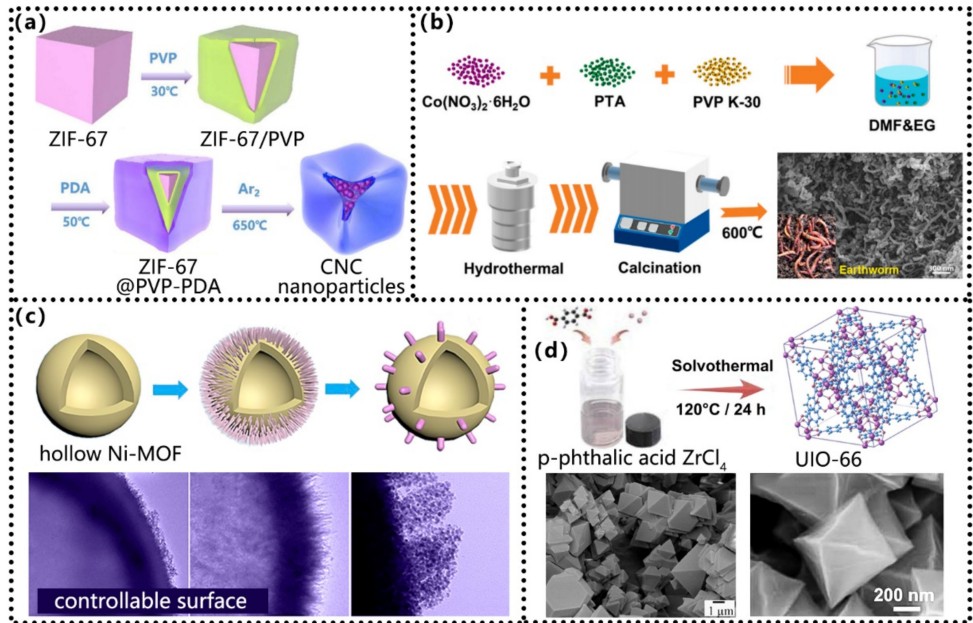

**Figure 7.** Preparation and morphology of single-metal MOFs microwave absorbing materials. (**a**) The synthesis process of CNC nanoparticles. Reprinted with permission from Ref. [82]. Copyright 2022, Elsevier. (**b**) The synthesis of earthworm-like (Co/CoO)@C composite. Reprinted with permission from Ref. [83]. Copyright 2021, Elsevier. (**c**) The synthesis of Ni-MOF hollow spheres with controllable surface architecture. Reprinted with permission from Ref. [84]. Copyright 2019, ACS Publications. (**d**) Diagram and SEM of UIO-66. SEM on the left: Reprinted with permission from Ref. [86]. Copyright 2014, Elsevier. SEM on the right: reproduced under the terms of the CC-BY Creative Commons Attribution 4.0 International license (https://linkspringer.53yu.com/article/10.1007/s408 20-021-00606-6 (accessed on 7 April 2022)) [85].

In order to improve the impedance mismatch of MOF-derived MAMs caused by sufficient permittivity and inadequate permeability, and to obtain broadband applications, the multi-metal MOFs integrating the advantages of multiple metals will be a desirable choice. Liu et al. [69] synthesized the CoNi/C nanocomposites derived from bimetallic MOFs. the obtained CoNi/C composites are composed of numerous fine particles with the size of 20 nm, and in the composites, Ni, Co, and C elements are uniformly distributed. The EM wave absorption mechanism for the CoNi/C sample is shown in Figure 8a. The CoNi/C sample with a high surface area can offer more multiple scattering sites. Additionally, the existence of carbon with abundant functional groups leads to both significant dipole polarization and effective conductive loss. The bimetallic metallic CoNi nanoparticles located on the carbon surface produce both magnetic loss and CoNi/C interfacial polarization loss.

The absorption properties of Zn-based MOF-derived MAMs are limited by the absence of magnetism. Therefore, few studies have been focused on pure Zn-contained MOFs derived MAMs. However, the introduction of magnetic components into Zn-containing MOFs and the combination of Zn and magnetic metals as hybrid centers to form multi-metal MOFs as MAM precursors have attracted considerable research interest in the field of microwave absorption. The percolation networks formed by the porous carbon will further strengthen the conductive loss. In reference [71], ZnCo-MOF was prepared by the wet chemical method, and porous Co/ZnO/C(CZC) was obtained after pyrolysis. When the calcination temperature is higher than 550 °C, the zinc element in the Zn-MOFs can be converted to ZnO. ZnO is a typical polarized semiconductor with low conductivity and a

wide band gap, which has been widely used to modulate microwave absorption properties. As the temperature rises further, the carbon reduced ZnO metal will start to evaporate, while the Co content increases, enhancing the magnetic loss. The multiple interfacial polarization between Co/C, ZnO/C, and C/paraffin increases the polarization loss, while the dipole polarization caused by many ZnO nanoparticles in the carbon layer generates a dielectric loss, as shown in Figure 8b. The improved absorption performance of multi-metal MOFs derivatives is related to the positive effects of band hybridization between elements of nanoparticles on dielectric constants and permeabilities [23]. As shown in Figure 8c, porous flower-like Ni/C composites can be prepared by pyrolysis of Zn-doped MOFs in a nitrogen atmosphere [87]. Due to the massive, porous and large spacing flakes in the 3D flower-like structure, the EM wave scattering is effectively increased.

Liu et al. [65] prepared porous hollow CNC(CoNi@C) microspheres derived from MOF through a facile solvothermal route and subsequent annealing process (Figure 8d). Due to the hollow structure, interface polarization and the synergistic effect between Co/Ni bimetallic and graphitized carbon, superior microwave absorption properties are obtained. Using MOFs as the precursor, Yang et al. [88] uniformly confined the highly dispersed Co and ZnO nanoparticles to the graphitized nitrogen-doped carbon framework and established a balanced electromagnetic performance in the absorbent. Meanwhile, the shape of MOF-derived metal/carbon-based materials was controlled by adding template agents and changing the metal source, and the transformation from a hexagonal star, flower, and cube to a random stone was realized (Figure 8e).

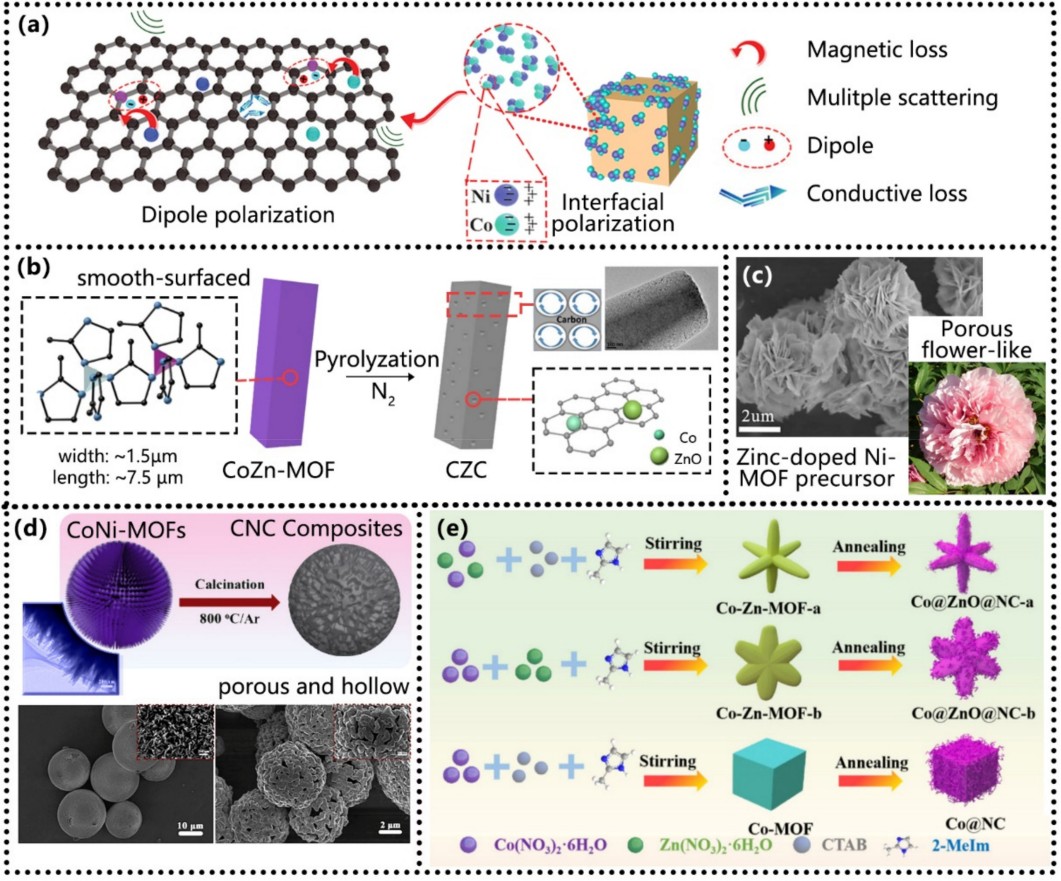

**Figure 8.** (**a**) Schematic diagram of the EM wave absorption mechanism for CoNi/C nanocomposites. (**b**) Porous CZC composites. Reprinted with permission from Ref. [71]. Copyright 2018, ACS Publications.

(**c**) Porous flower-like Zinc-doped Ni-MOF precursor. Reprinted with permission from Ref. [87]. Copyright 2019, Elsevier. (**d**) The synthetic process of hollow CNC microspheres and SEM images of CoNi-MOFs and CNC-1:1. Reprinted with permission from Ref. [65]. Copyright 2019, Elsevier. (**e**) The synthesis process and formation mechanism of Co@ZnO@NC-a, Co@ZnO@NC-b, Co@NC. Reprinted with permission from Ref. [88]. Copyright 2022, Elsevier.

### 4.2. Compound Materials Based on MOFs

Compounding MOF with one-dimensional (Fibrous), two-dimensional (Flake), and three-dimensional (spherical) materials can not only balance electromagnetic parameters, but also controls the growth of MOF on different frames, avoids irregular dispersion or agglomeration of particles, and further improves the microwave absorption performance of the material.

**One-dimensional composites.** Compared with spherical particles, non-spherical particles have anisotropic and special physical and chemical properties. One-dimensional nanomaterials have a large aspect ratio, which can provide a longer transmission channel for dissipative current and is conducive to electrical loss. Due to the lightweight and great EM wave attenuation characteristics of carbon fiber and MOF derivatives, MOF-derived/carbon fiber composite materials have great potential as excellent absorbing materials. Chen et al. [89] prepared $Fe^{III}$-MOF-5-derived/carbon fibers composites (FMCFs) through facile electrospinning and heat treatment strategy (Figure 9a). The great EM absorption performance of FMCFs is due to the high-density magnetic flux line around the magnetic particles in the carbon fiber, which will affect the external electromagnetic field to obtain magnetic coupling [90]. Secondly, the interface composed of the microstructure of porous $Zn/Fe/Fe_3O_4$/carbon fiber makes the positive and negative charges distributed around the contact area. Therefore, the dynamic balance of migration and diffusion can promote the formation of local dipole moment fields, and the charge distribution can be well-organized along the propagation direction of the EM wave. In the process of polarization and relaxation, the EM wave energy can be well attenuated in the dynamic balance [90,91]. In addition, under the action of the electric field, carbon fiber can form a conductive current, and the 3D conductive network established by FMCFs units can also effectively enhance the EM absorption performance of the material [91].

**Two-dimensional composites.** Zhang et al. [92] used EGaIn as an aluminum reservoir to prepare layered double hydroxide (LDH) and MOFs nano-arrays. The prepared CoAl-LDH@ZIF 67 can be transformed into CoAl-LDO@Co-C in the subsequent annealing process performed under nitrogen environments. As a conductive carbon material, GO has a large number of active sites on its surface, which can adsorb metal ions through electrostatic interaction to form MOFs in situ. The SEM image in Figure 9b is the MOF/GO hybrid prepared by Zhang et al. [66]. It can be seen that MOF is scattered on the surface of GO with an average size of several hundred nanometers and appears as rhombic dodecahedral morphology. Yuan et al. [72] also synthesized carbonaceous $Co_3O_4$/Co/RGO composites as high-performance MAMs with cobalt-based ZIF-67 and GO as precursors. The derived carbonaceous $Co_3O_4$/Co microframes retain the structure of ZIF-67 nanoparticles and are well dispersed onto the surface of RGO nanosheets. As shown in Figure 9b, ZIF-67@CoNi LDHs-GO was used as a precursor to prepare a hollow granatohedron structure consisting of CoNi nanoparticles embedded within nanoporous N-doped carbon polyhedrons grown onto rGO (CoNi@NCPs-rGO) [93].

**Three-dimensional composites.** Zhang et al. [94] synthesized a Co-Fealloy@N-Doped carbon hollow spheres by dual-MOF-assisted pyrolysis approach (Figure 9c). After thermal treatment, the discrete Fe-based MOF nanocrystallites can form hollow composites composed of Co-Fe alloy nanoparticles homogeneously distributed in porous N-doped carbon nanoshells. In reference [95], $NiFe_2S_4$/PC composites with excellent properties were prepared by introducing Ni-MOF, Fe, and S elements into a three-dimensional porous carbon framework (Figure 9d). Song et al. [96] used MOF and micro-sized PG networks as precursors to synthesize a novel porous $ZnO/ZnFe_2O_4$/C@PG composites through a synchronous

reflux strategy (Figure 9e). As a multi-component composite, the $ZnO/ZnFe_2O_4/C$ component provides excellent dielectric loss and certain magnetic loss, while the PG conductive network has the strongest dielectric loss ability due to its high surface conductivity and rich functional groups, which can produce more electron transmission paths and significant conductive loss. Among them, PG has good EM absorption performance at low frequencies, and the composite $ZnO/ZnFe_2O_4/C@PG$ has the obvious effect of broadening the frequency band. Tao et al. [97] used lightweight and conductive CF as the core and cubic ZIF-67-derived C/Co with controllable structure and adjustable graphitization as the shell to prepare a series of core–shell CF@C/Co composites (Figure 9f). After properly adjusting the degree of graphitization on the basis of derivation of various C/Co shells, A4-700 with good EM absorption performance at low frequencies can be obtained.

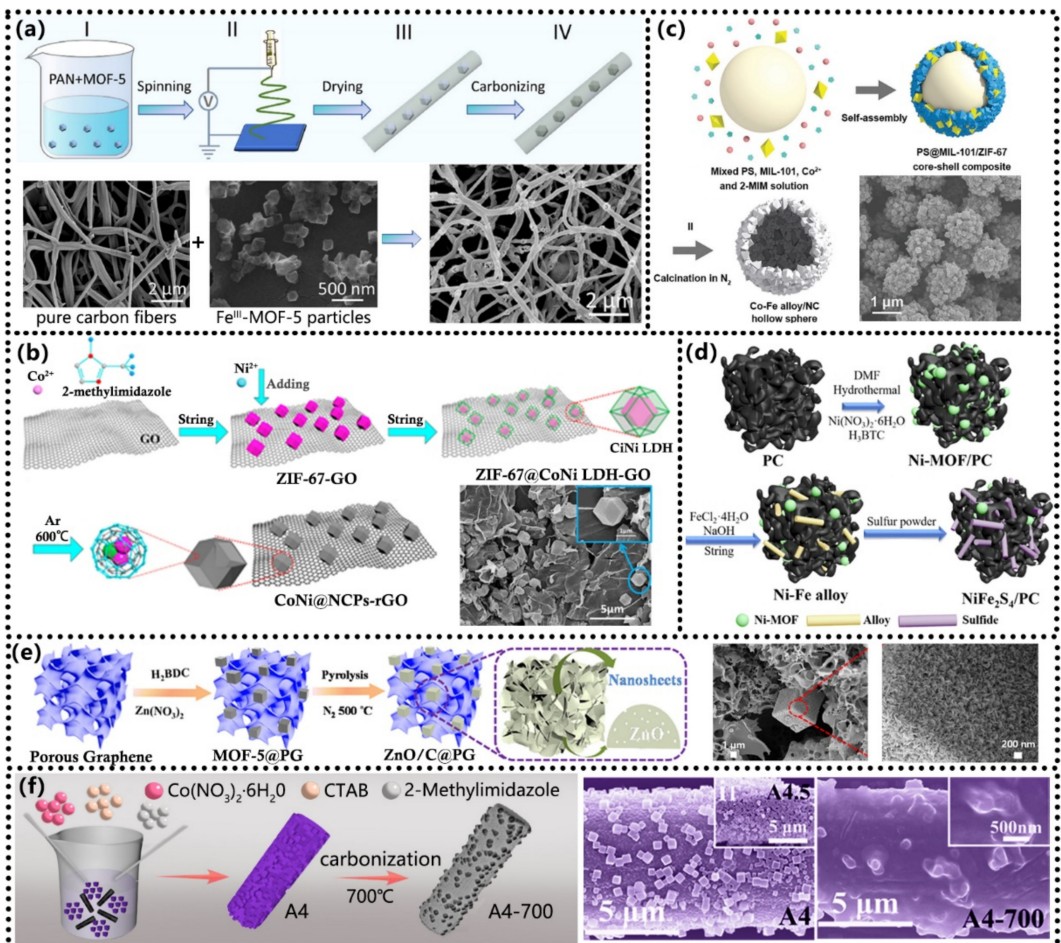

**Figure 9.** (**a**) The synthesis strategy of FMCFs composite. Adapted with permission from Ref. [89]. Copyright 2021, Elsevier. (**b**) The synthesis process of CoNi@NCPs-rGO. Adapted with permission from Ref. [93]. Copyright 2021, Elsevier, and the SEM images of MOF/GO hybrid. Reprinted with permission from Ref. [66]. Copyright 2017, Elsevier. (**c**) The formation process of Co–Fe alloy/N-doped carbon hollow spheres and FESEM images of Co–Fe/NC-700 hollow spheres. Adapted with permission from Ref. [94]. Copyright 2019, John Wiley and Sons. (**d**) The synthesis process of $NiFe_2S_4/PC$. Adapted with permission from Ref. [95]. Copyright 2022, Elsevier. (**e**) The synthesis process of ZnO/C@PG. Adapted with permission from Ref. [96]. Copyright 2021, Elsevier. (**f**) The synthesis of CF@ZIF-67 and CF@C/Co (carbonized at 700 °C) and their SEM images. Adapted with permission from Ref. [97]. Copyright 2021, Elsevier.

**Composites of MOFs.** Constructing multiple MOF-on-MOF heterostructures is challenging. Wu et al. [98,99] constructed the binary component DUT-52@MIL-88B (DM) by combining two unique anisotropic epitaxial growth strategies (Figure 10a). As shown in Figure 10(a$_2$), the SEM image of DM shows a special hybrid structure of hexagonal pyramid column and octahedron. The formation of this structure is due to the selective growth of the guest MOF on the specific crystal facet of the host MOF. DUT-52@MIL-88B@MIL-88C (DMM) is formed by the ternary assembly. They can be converted into magnetic porous carbon-based absorbers (DM-700 and DMM-700) through a facile carbonization process. Compared with DM-700, the optimized absorber DMM-700 has an obvious effect of broadening the frequency band. As shown in Figure 10b, core–shell structure and bimetallic composition Cu/NC@Co/NC composites were designed and synthesized through the thermal decomposition of Cu-MOF@Co-MOF precursor [100]. Compared with the Cu/NC and Co/NC, the Cu/NC@Co/NC composite exhibits many excellent EM absorbing performances caused by the bimetallic composition and the unique core–shell structure. Peng et al. [101] prepared a series of Co, metal oxide semiconductor, and NPC composites with a multi-nut-bread structure by the one-step pyrolysis of capsule MOFs@MOFs (Figure 10c). The impedance matching and attenuation ability of the absorbers can be adjusted by tuning the type and content of "nuts" with different electromagnetic characteristics.

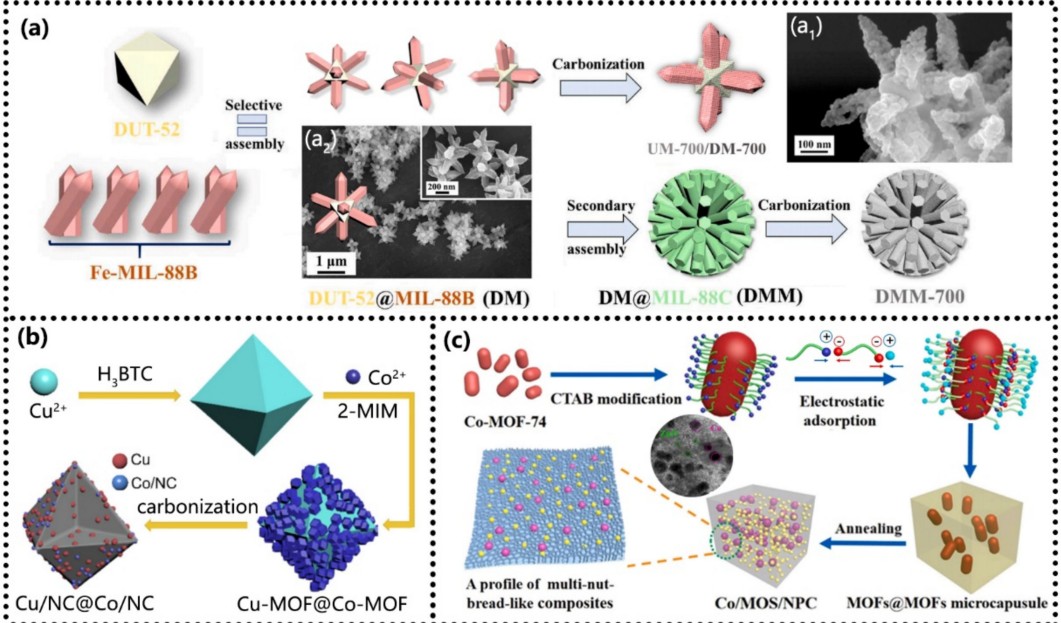

**Figure 10.** (a$_1$) The synthetic process of UM-700 and DM-700, and the XRD patterns of DM- 700. Adapted with permission from Ref. [99]. Copyright 2021, Elsevier. (a$_2$) The synthetic process of DMM-700 and the SEM image of DM. Adapted with permission from Ref. [99]. Copyright 2022, Elsevier. (b) The synthetic process of Cu/NC@Co/NC composites. Adapted with permission from Ref. [100]. Copyright 2022, Elsevier. (c) Synthetic process of Co/MOS/NPC composites. Adapted with permission from Ref. [101]. Copyright 2021, Elsevier.

In addition, the MOF core–shell structure can also be constructed to improve the interface polarization by adjusting the micro regulatory factors (such as the size and thickness of the core and shell, the structure of the material (heterogeneous interface, crystal defects, pore structure) the thickness of the absorbing layer and other factors), so as to further improve the EM absorption performance of the materials [102].

As shown in Figure 11a, bimetallic CoFe-MOF@Ti$_3$C$_2$T$_x$ MXene derived multiple interfacial composites with controllable structures were prepared by solvothermal, electrostatic self-assembly, and thermal treatment [103]. The synthesized core–shelled CoFe/C@TiO$_2$/C

composite exhibits excellent EM absorption properties. As shown in Figure 11b, Li et al. [104] synthesized hollow structure Z67-8@C through a simple preparation method. Rösler et al. [105] demonstrated the formation of well-defined hollow Zn/Co-based ZIFs by the use of epitaxial growth of ZIF-8 on preformed Co-MOF ZIF-67 nanocrystals that involve in situ self-sacrifice/excavation of the Co-MOF (Figure 11($c_1$)). Hollow MOF has adjustable porosity and a multifunctional structure, and any type of metal nanoparticles can be accommodated in Zn/Co-ZIF shells to generate yolk–shell metal@ZIF structures. Therefore, it has broad development prospects in the field of EM absorption. Cui et al. [106] used the bimetallic core–shell structure ZIF-67@ZIF-8 is used as the template, $Ni(NO_3)_2·6H_2O$ is used as the etchant, and the heterogeneous trimetal Co@ZnO/Ni@NC nanocage is prepared by Ni doping and vacuum carbonization. The nanocage uses Co as the core and ZnO/Ni particles coated with a nitrogen-doped carbon layer as the shell layer (Figure 11($c_2$)). Wang et al. [107] prepared a novel hierarchical multi-interfacial Ni@C@ZnO microsphere with a special Schottky contact structure by annealing the bimetallic Ni-Zn-MOF precursor. As shown in Figure 11d, the unique yolk–shell microspheres were assembled together by the core–shell Ni@C micro-units and ZnO flakes. Additionally, the ZnO flakes were anchored inside the hierarchical conductive carbon matrix uniformly. The FeCoNi MOF prepared by the hydrothermal method has a core–shell structure (FeCoNi@C) after pyrolysis [70] (Figure 11e). The hysteresis loss and magnetic resonance of biomagnetic metals lead to the magnetic loss of the composites, and the doping of Fe further increases the hysteresis loss. On the other hand, the absorbed EM wave is reflected and scattered in the hollow porous structure, which greatly enhances the interface polarization and dielectric loss of the material. FeCoNi-MOF-700 (sintering at 700 °C) with a thickness of 4.74 mm has the best comprehensive performance, with the $RL_{min}$ of −69.03 dB at 5.52 GHz and an EAB of 2.5 GHz.

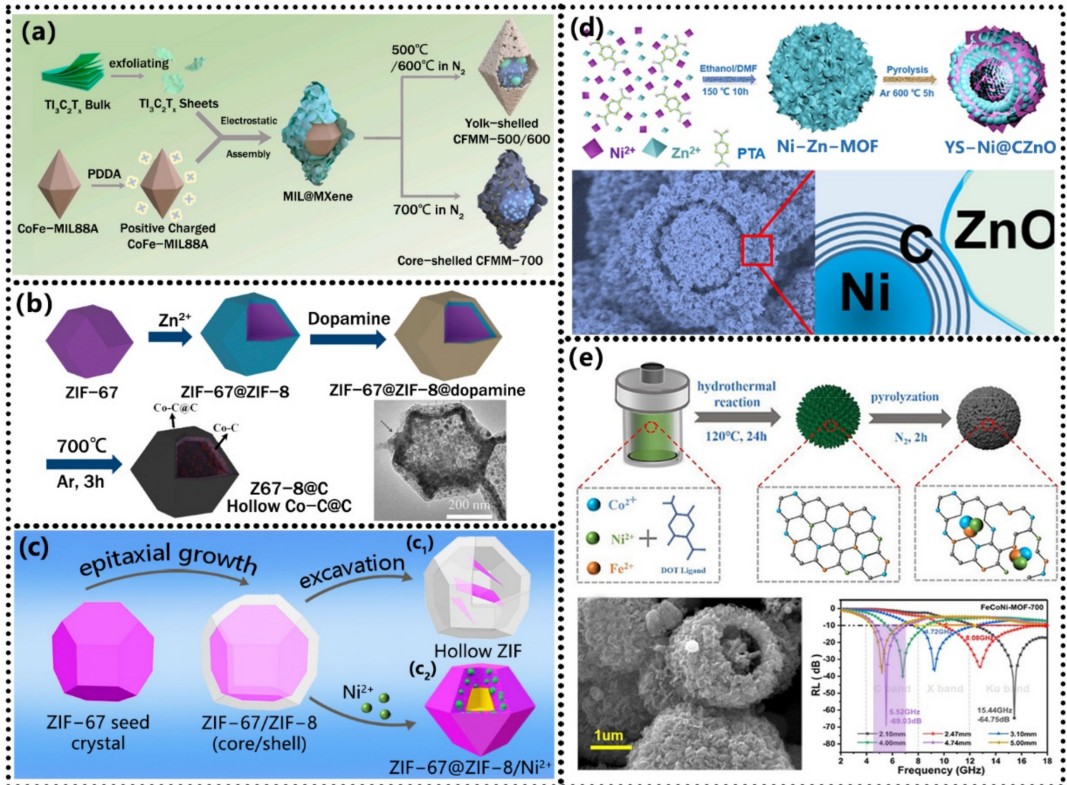

**Figure 11.** (**a**) The synthesis process of CFMM. Adapted with permission from Ref. [103]. Copyright 2022, Elsevier. (**b**) The synthesis of ZIF67-8@C and TEM image of ZIF67-8@C. Adapted with permission from Ref. [106]. Copyright 2021, Elsevier. (**c₁**) The synthesis process of hollow ZIF. Adapted with permission from Ref. [105]. Copyright 2016, John Wiley and Sons. (**c₂**) ZIF-67@ZIF8/Ni²⁺. Adapted with

permission from Ref. [104]. Copyright 2021, Elsevier. (**d**) The synthetic process of yolk–shell Ni@C@ZnO microspheres. Adapted with permission from Ref. [107]. Copyright 2020, Elsevier. (**e**) The synthesis procedure of the FeCoNi@C nanocomposites and the SEM image of FeCoNi-MOF-900, *RL* of FeCoNi-MOF-700. Adapted with permission from Ref. [70]. Copyright 2019, American Chemical Society.

## 5. Process Controls in Preparation

### 5.1. Sintering Temperatures Control

Previous studies have demonstrated that porous nanostructures of metal, metal oxides, carbon, or hybrids can be converted from MOF precursors by thermal decomposition, depending on calcination conditions, (e.g., atmosphere, temperature, and time) [108,109]. Figure 12a shows the TG curves of ZIF-67 precursors recorded under air and inert gas ($N_2$). By comparison of the two curves, the total weight loss in the air is very close to the theoretical value of ZIF-67 into $Co_3O_4$. However, in $N_2$, the total weight loss value is significantly lower than the theoretical value of ZIF-67 conversion to metal cobalt or its oxide. This result implies that during the thermal decomposition of MOFs under an inert atmosphere, organic components of MOFs can be directly carbonized, and metal ions are reduced in situ, thereby forming metal/carbon hybrids. However, even a small amount of oxygen in the atmosphere during thermal decomposition would lead to the preferential formation of metal oxides in the products. To avoid this situation, thermal decomposition should be carried out in a high-purity atmosphere as far as possible.

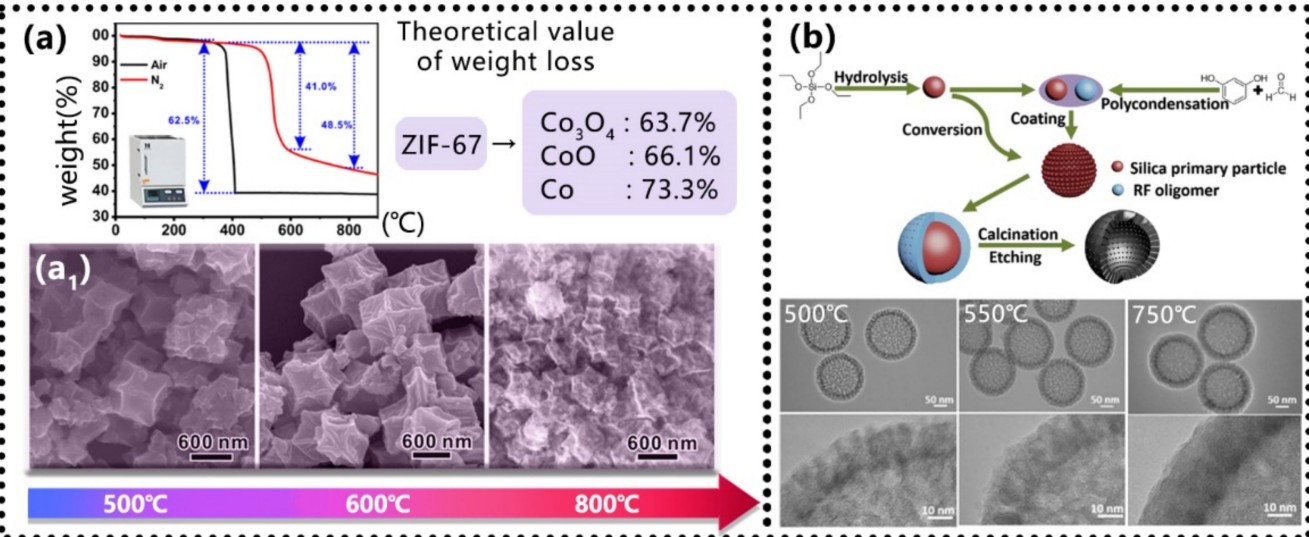

**Figure 12.** (**a**) TG curves of the Co-MOF (ZIF-67) precursors under air and $N_2$ and (**a₁**) SEM images of Co/C-500, Co/C-600, and Co/C-800. Adapted with permission from Ref. [15]. Copyright 2015, American Chemical Society Publications (**b**) The formation of PCHMs and SEM images of PCHMs (carbonized at 500 °C, 550 °C, and 750 °C). Adapted with permission from Ref. [110]. Copyright 2019, Elsevier.

The pyrolysis temperature mainly affects the graphitization degree of carbon matrix and the microstructure of MAMs. The graphitization degree is positively correlated with pyrolysis temperature; a higher graphitization degree can provide greater electrical conductivity, which is conducive to the conductivity loss of incident microwave. In the process of pyrolysis, a large number of oxygen vacancies and defects at the boundary of the lower graphitized carbon layer are also conducive to the enhancement of impedance matching and polarization relaxation. Liu et al. [15] obtained three kinds of Co/C composites at different temperatures. Through the analysis of the experimental results, it is found that the saturation magnetization and coercivity of Co/C composites increase with the increase in

calcination temperature. This is mainly because the crystallinity of cobalt particles increases with the increase in calcination temperature [55]. However, this does not mean that the higher the sintering temperature, the better the microwave absorption performance. When the EM wave is incident on the surface of MAMs, the induced currents are stimulated, leading to enhanced reflection, and weaker absorption, resulting in worse microwave absorption performance, which is called the "skin effect". On the other hand, surplus conductivity can interact with the incident microwave, to form an induced magnetic field in the high-frequency magnetic field, resulting in magnetic energy radiation and negative magnetic loss. In addition, the ultra-high pyrolysis temperature might lead to the loss of organic matter, the surface depression of magnetic nanoparticles, the collapse of MOF skeletons, and the smaller gap between particles, which are more prone to agglomeration (as shown in Figure 12($a_1$)), which is not conducive to multiple reflections and electromagnetic synergy. Among these, Co/C-500 obtained at 500 °C showed the best EM wave absorption performance. In particular, when the frequency is 5.8 GHz and the thickness is 4 mm, $RL_{min}$ of Co/C-500 reached −35.5 dB and the EAB was 2.3 GHz (4.8–7.1 GHz).

In addition, the pore size and shell thickness of particles can also be changed by adjusting the pyrolysis temperature. Qingdao University [110] prepared carbon hollow microspheres with uniform mesoporous shell structure (PCHMs) by template assisted method. By adjusting the pyrolysis temperature, PCHM with different pore sizes and shell thicknesses were obtained, as shown in Figure 12b. According to the waveguide theory, when the wavelength is twice the length of the waveguide cross-section, the microwave can be dissipated in the waveguide. Therefore, different pore sizes correspond to different microwave lengths, and the gradient distribution of the pore structure is also conducive to broadening the EAB.

*5.2. Materials Ratios Control*

By adjusting the molar ratio of raw materials, MAMs with different core–shell thicknesses, different pore sizes, different interface structures, and even different morphology can be generated, so as to adjust the electromagnetic parameters and achieve the purpose of low-frequency and wide-band EM absorption.

Liu et al. [111] successfully prepared one-dimensional sponge metal Co with strong magnetism at low frequencies through direct pyrolysis of $Co_3[HCOO]_6 \cdot DMF$ parallelepiped and prepared Co/C composites with a unique interface structure. With an increase in the carbon content, the morphology of carbon changes from fragments to vertically arranged nanosheets, as shown in Figure 13a (S-Co/C-x, x means the weight of glucose that provides the carbon source). Jin et al. [112] prepared the film with controllable thickness by controlling the amount of pyridine added to Ni@NC Hexagonal nanosheets, whose morphology can be adjusted from nanorods, and nanoparticles to nanosheets with the increase in the doping content of N atoms (Figure 13b). The 3D conductive network of carbon framework and magnetic Ni nanoparticles could be obtained from the converted MOF precursor. The controllable dimension and morphology of the N-doped carbon skeleton endowed more electronic transportation paths, better anti-reflection surfaces, higher conduction loss, and polarization relaxation, while the evenly distributed Ni nanoparticles provided considerable multiple resonances and eddy current. Kong et al. [113] prepared the $Fe_2O_3$@ZnCo-MOF composites by the in situ growth method. By varying the loading of $Fe_2O_3$ during the synthesis process, a series of composites were obtained. Among them, CoFe alloys@ZnO@C-0.1 (Figure 13c) can reach the $RL$ of −40.63 dB at a thickness of 2.2 mm, and the EAB is 3.8 GHz (4–7.8 GHz).

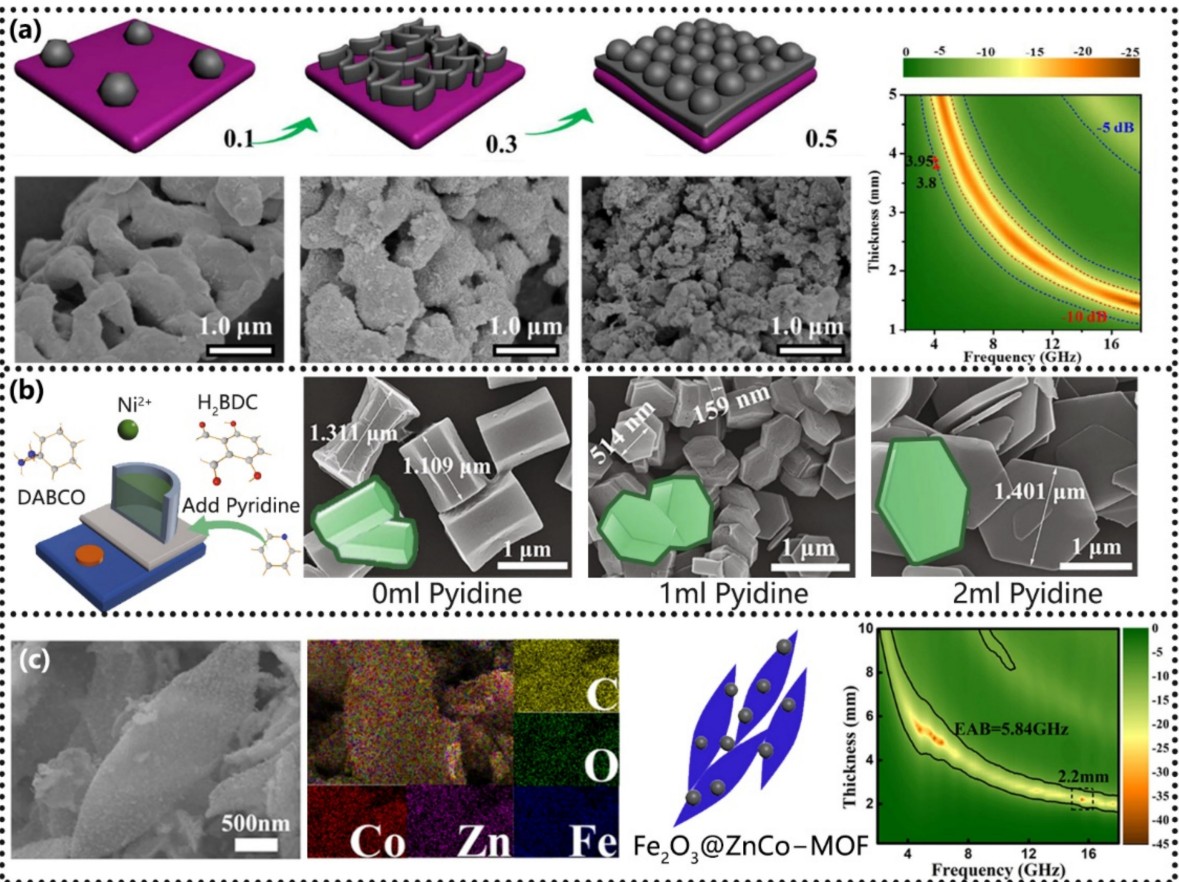

**Figure 13.** (**a**) S-Co/C-0.1, S-Co/C-0.3, and S-Co/C-0.5. Adapted with permission from Ref. [111]. Copyright 2018, American Chemical Society; (**b**) Schematic preparation process of Ni@NC composites and the SEM images of the precursors with different dosages of pyridine. Adapted with permission from Ref. [112]. Copyright 2022, Elsevier. (**c**) SEM images and elemental mappings of CFZC-0.1. Adapted with permission from Ref. [113]. Copyright 2021, Elsevier.

Wu et al. [114] constructed two MOFs composite nanoflowers with anisotropic binary assembly structures by a facile one-pot method. As shown in Figure 14($a_1$–$a_{12}$), by changing the ratio of Fe ions to Co ions or Ni ions, the morphology control of the MOFs was successfully achieved. The transformation behaviors of the morphology could be attributed to the changes in the specific proportion of the MIL-88B and MOF-5 structures in the binary assembled structure along with the ion ratios. The results indicated that the production of the binary assembled nanoflower structure was the result of the coordinated action of the three metal ions: Fe ion, Co ion, and Ni ion. By controlling the solvent and molar ratio of cobalt/linker, as shown in Figure 14b, Huang's research group made the multidimensional controllable MOFs derivatives nitrogen-doped carbon materials exhibit tunable morphology from sheet-, flower-, cube-, dodecahedron- to octahedron-like After pyrolysis, Co@NC-MOF composites have uniform charge density distributions, rich interfacial polarization, and strong magnetic coupling networks. Meanwhile, the dielectric carbon skeleton of MOF derivatives also provides electron transport paths and enhances conductive dissipations. Among them, Co@NC-c and Co@NC-o have good absorption performance at low frequencies. Co@NC-c with a thickness of 3.5 mm has an EAB of 2.25 GHz (5.5–7.75 GHz) and a minimum reflection loss of −30 dB. Co@NC-o with a thickness of 4 mm has an EAB of 2 GHz (6.75–4.75 GHz) and a minimum reflection loss of −40 dB [115].

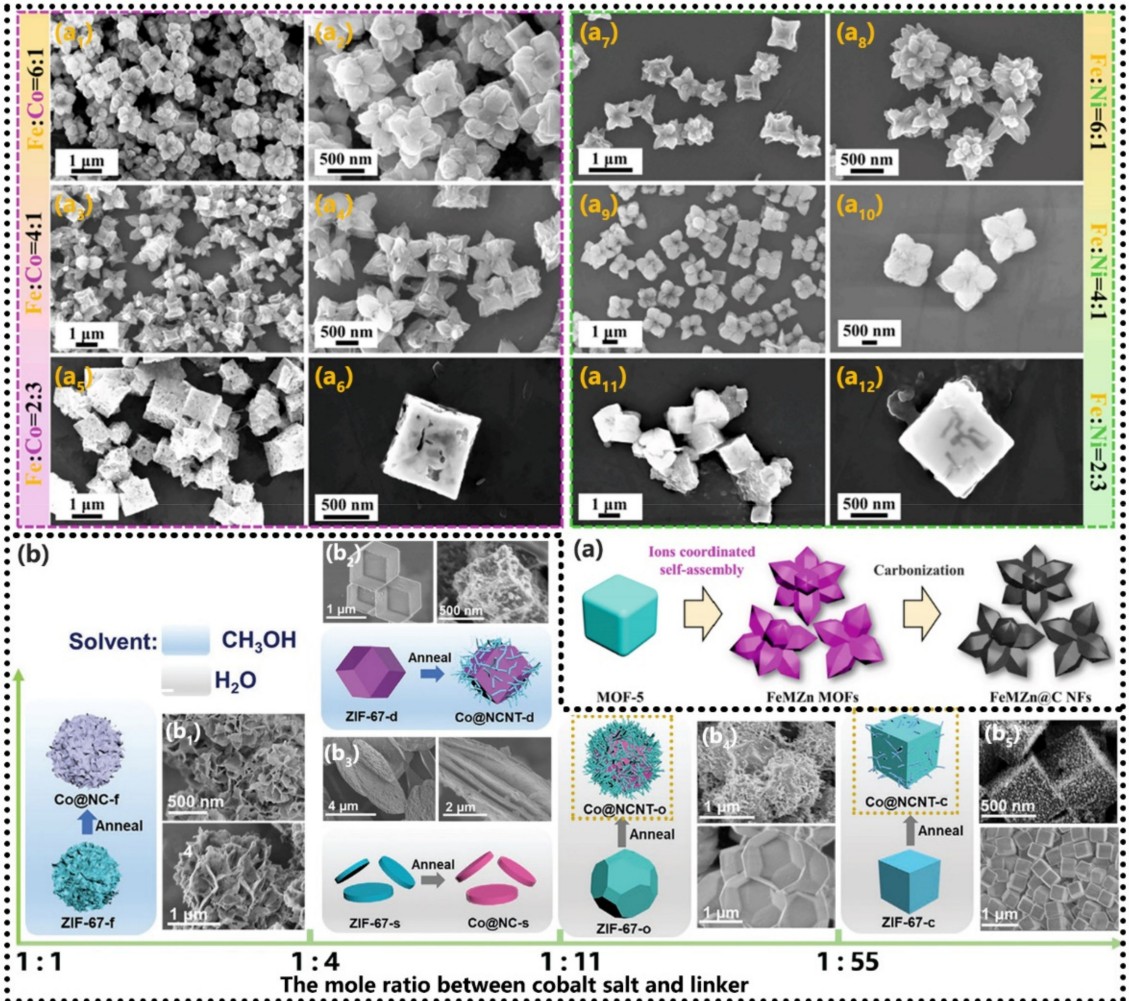

**Figure 14.** (**a**) The synthesis process of FeMZn-MOFs and SEM images of FeCoZn MOFs: Fe:Co = 6:1 (**a₁,a₂**), Fe:Co = 4:1 (**a₃,a₄**) and Fe Co = 2:3 (**a₅,a₆**); FeNiZn MOFs: Fe:Ni = 6:1 (**a₇,a₈**), Fe: Ni = 4:1 (**a₉,a₁₀**) and Fe:Ni = 2:3 (**a₁₁,a₁₂**). Adapted with permission from Ref. [114]. Copyright 2022, Elsevier. (**b**) The synthesis process of Co@NC composites and SEM images of Co@NC-f with ZIF-67-f as the precursor (**b₁**), Co@NCNT-d with ZIF-67-d as the precursor (**b₂**), Co@NC-s with ZIF-67-s as the precursor (**b₃**), Co@NCNT-o with ZIF-67-o as the precursor (**b₄**), Co@NCNT-c with ZIF-67-c as the precursor (**b₅**). Adapted with permission from Ref. [115]. Copyright 2020, Small.

## 6. Challenges and Prospects

This paper summarized the research progress of absorbing composites based on MOF derivatives, and the factors that affect the absorbing performance at low frequencies. With the deepening of research, researchers have gained a comprehensive understanding of MOF derivatives as efficient MAM, and have made reasonable assumptions about the reaction mechanism during the treatment process through various characterization methods. However, the materials that satisfied both the basic requirements of "thin, light, broad, strong" and broadband absorption in low frequencies are still to be discovered.

Figure 15 summarizes the EAB, $RL_{min,}$ and the corresponding thickness of the MAMs with good EM absorption performance at low frequencies in the literature. It can be seen that the EAB at low frequencies is limited (the EAB rarely exceeds 3 GHz). Most MAMs can achieve effective wave absorption in the C-band, but the frequency band still needs to be broadened through material selection, interface design, material compound, and so on. Relatively few MAMs can achieve wave absorption in L-band and S-band. For

the follow-up research on "low-frequency broadband" MAMs, the following aspects can be considered:

- Design and selection of composites. At present, MOF derivatives applied in the field of absorption can be divided into single-metal derivatives, multi-metal derivatives, oxide/sulfide derivatives, and other composite materials according to their compositions. The metal atoms mainly include Fe, Co, Ni, and Zn. Subsequent research on other metal elements should be increased, and combinations of different metals should be tried to achieve synergistic effects. Meanwhile, we should also pay attention to the ionic potential of metals combined with ligands and explore its corresponding relationship with low-frequency absorption ability [22]. In addition, MOFs products are also closely related to the ratio between metals and organic ligands, which should be recorded as their volume ratios for the ease of repeated experiments and large-scale productions.

- Structural optimization. After different treatment processes, MOF precursors can be compounded with one-dimensional and two-dimensional materials. One-dimensional MOF derivatives have large specific surface areas due to their large length and diameter. Two-dimensional MOFs derivatives have a network structure, which allows multiple reflections of electromagnetic waves inside and can effectively improve the attenuation ability. In addition, the application of MOF is not limited to the field of microwave absorption, MOFs@MOFs, MOFs of core–shell structure and multilayer core–shell structure, CelloMOFs [116], Zr-MOF [117], MOF/COF-based hybrid nanomaterials [118] and other MOF materials are also widely used in water treatment, air purification, biology, energy storage, catalysts, membrane separation [116–119], supercapacitors [116,120] and other fields that have broad development prospects.

- Controlled preparation. With the aim of obtaining materials with the "low-frequency broadband", comparative experiments should be conducted with variables of different materials (metal and organic ligand), material ratios, proportions of the dissolved agent, pyrolysis temperatures, and environment, and paraffin filling amounts.

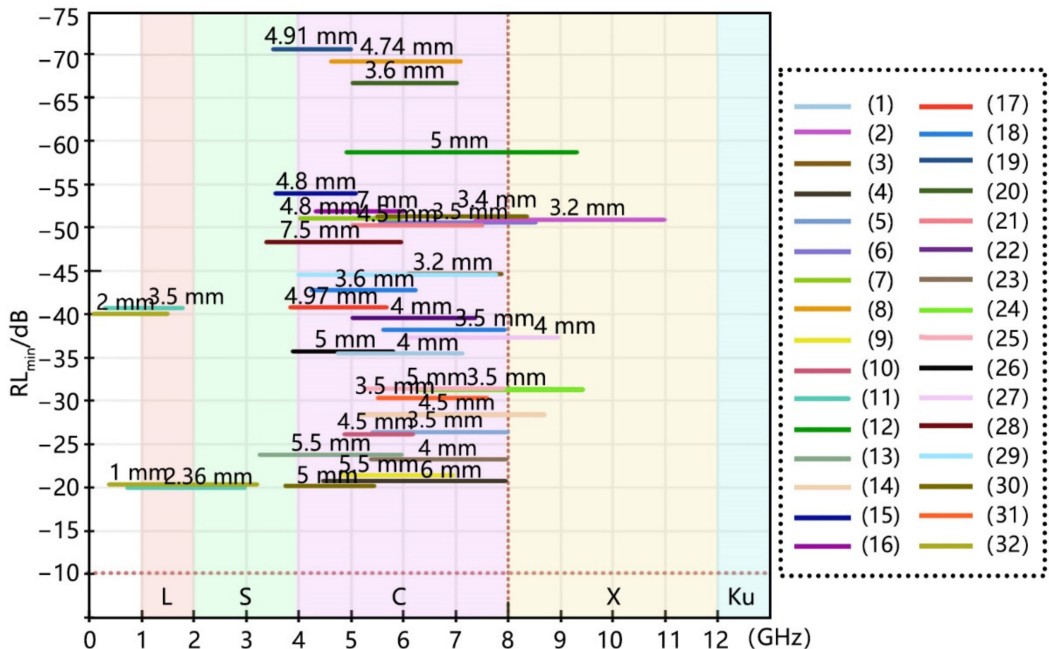

**Figure 15.** Comparison of EM wave absorption properties of MOF absorbing materials. (1) data from Ref. [15]; (2) data from Ref. [64]; (3) data from Ref. [65]; (4) data from Ref. [66]; (5) data from Ref. [67]; (6) data from Ref. [68]; (7) data from Ref. [69]; (8) data from Ref. [70]; (9) data from Ref. [71]; (10) data

from Ref. [82]; (11) data from Ref. [83]; (12) data from Ref. [84]; (13) data from Ref. [85]; (14) data from Ref. [87]; (15) data from Ref. [88]; (16) data from Ref. [92]; (17) data from Ref. [93]; (18) data from Ref. [96]; (19) data from Ref. [97]; (20) data from Ref. [99]; (21) data from Ref. [100]; (22) data from Ref. [101]; (23) data from Ref. [103]; (24) data from Ref. [104]; (25) data from Ref. [106]; (26) data from Ref. [107]; (27) data from Ref. [110]; (28) data from Ref. [112]; (29) data from Ref. [113]; (30) data from Ref. [114]; (31) data from Ref. [115]; (32) data from Ref. [121].

## 7. Concluding Remarks

This paper summarizes a variety of methods to realize low-frequency and broadband EM wave absorption by changing the composition, preparation methods, and conditions of MOF derivatives. The main takeaways are summarized below:

- In order to realize effective absorption of low-frequency waves, the impedance matching at low frequencies should be realized, so that the electromagnetic wave can enter the MAM to realize the energy conversion. Because the way of EM wave loss in a good conductor is mainly magnetic loss and the value of the permeability of the absorber is usually much smaller than the dielectric constant, the magnetic loss is more important than the dielectric loss. The low-frequency impedance matching can be improved by adding magnetic materials.
- In order to make full use of magnetic loss materials to effectively attenuate EM waves, nanostructured MOFs materials that meet the critical radius of single domains and produce resonance absorption size can be added to the polymer matrix, composed of low-dimensional or multi-dimensional materials, so as to produce excellent microwave absorption properties.
- In order to realize multiple reflections and interface polarization, the material size should be controlled at the nanometer level. Due to the large specific surface area of nanomaterials, attention should also be paid to the treatment of the interface, because agglomeration (cluster) will lead to the decrease in dielectric constants.
- When preparing MOF materials, a variety of metal ions can be compounded. In addition, a variety of materials with good microwave absorbing properties in different frequencies can be mixed. In this way, the microwave absorbing properties of overlapping parts can be enhanced and the broadband can be broadened.
- The sintering temperature and the matching thickness in the preparation process are also very important. At present, researchers are still not very clear about the reaction principle in the material preparation process, where there are many influencing factors; therefore, the comparative tests aiming at low frequencies and broadband are extremely critical.

**Author Contributions:** Writing—original draft preparation, W.S.; writing—review and editing, Q.L., W.H., L.C., X.L., Y.S., Z.Z., M.S. and L.Q.; supervision, Q.L., L.C., Y.S. and L.Q. All authors have read and agreed to the published version of the manuscript.

**Funding:** This work was supported by the National Natural Science Foundation of China (Nos. U2006218, 901203520, 51971029, 61871043, 62101055), BRICS STI Framework Program by NSFC (No. 51861145309), Qin Xin Talents Cultivation Program, Beijing Information Science and Technology University (QXTCP A202103) and Scientific research level improvement project–key research cultivation project, Beijing Information Science and Technology University (2020KYNH221).

**Institutional Review Board Statement:** Not applicable.

**Informed Consent Statement:** Not applicable.

**Data Availability Statement:** Not applicable.

**Conflicts of Interest:** The authors have no conflicts of interest in the design of the study; in the collection, analyses, or interpretation of data; in the writing of the manuscript, or in the decision to publish the results.

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
