# Peer review of "Low-Frequency Broadband Absorbing Coatings Based on MOFs: Design, Fabrication, Microstructure and Properties"

_coatings, doi:10.3390/coatings12060766_

Round 1

Reviewer 1 Report

Dear Authors,

Since I had recommended to accept the previous version of the manuscript before, after your careful improvement and taking into account my own comments, and now I can see the newer enhancement of the text, I believe it can be suitable for publication in Coatings in its present form.

Author Response

First, we would like to express our thanks to the reviewers for their instructive comments concerning our manuscript entitled ‘Low Frequency Broadband Absorbing Coatings Based on MOFs: Design, Fabrication, Microstructure and Properties’ (coatings-1575837). We have studied the comments carefully, and then carefully revised the whole manuscript. The revisions/ explanation corresponding to the comment are shown in the following (Reviewers' comments are in italic font):

Reviewer #1: Since I had recommended to accept the previous version of the manuscript before, after your careful improvement and taking into account my own comments, and now I can see the newer enhancement of the text, I believe it can be suitable for publication in Coatings in its present form.

Reviewer #2: L. Qin and co-workers revised and enhanced considerably their proposed review, now entitled Low Frequency Broadband Absorbing Coatings Based on MOFs: Design, Fabrication, Microstructure and Properties as I suggested. Indeed, almost all my suggestions were attended in satisfactory manner. For instance, all figures have now enough sharpness, a new conclusion remarks section apart from Challenges and Prospects section was included, and as I mentioned before, the tittle was modified as I suggested. For these reasons, I recommend publication of this manuscript in its present form.

Reviewer #3: The article contains 128 literary references, which testifies to the great work of the authors on the analysis of data accumulated in the world on the creation and use of materials that absorb microwaves.

Given the quality and nature of the data presented, I believe that this article will be of interest to a wide audience in this field and can be published without significant changes.

Reviewer #4: The authors reviewed applications of MOFs as adsorbents for low-Frequency Broadband. The submission is accepted for publication after considering the following points:

A: We are really grateful for the comments from the reviewer, and after accepting your suggestion, we have revised the following contents of the paper:

  1. An in-depth discussion regarding the topic should be included.

A: In order to have a more in-depth discussion on the topic and explore the mechanism, we have added new contents in different paragraphs. The content has been revised as:

(Page 9. Line 298): Added “The defects produced in N atom and carbon matrix can be used as polarization centers to introduce more dipole polarization and enhance the EM wave absorption properties.

(Page 10. Line 307): Added “improve the impedance mismatch of MOF-derived MAMs caused by sufficient permittivity and inadequate permeability, and to

(Page 10. Line 319): Added “When the calcination temperature is higher than 550°C, the zinc element in the Zn-MOFs can be converted to ZnO. ZnO is a typical polarized semiconductor with low conductivity and wide band gap, which has been widely used to modulate microwave absorption properties. As the temperature rises further, the carbon reduced ZnO metal will start to evaporate, while the Co content increases, enhancing the magnetic loss.

(Page 11. Line 352): Added “Compared with spherical particles, non-spherical particles have anisotropic and special physical and chemical properties. One dimensional nano materials have a large aspect ratio, which can provide a longer transmission channel for dissipative current and is conducive to electrical loss.

(Page 12. Line 372): Added “As a conductive carbon material, GO has a large number of active sites in its surface, which can adsorb metal ions through electrostatic interaction to form MOFs in situ.

(Page 14. Line 464): The absorbed EM waves is reflected and scattered in the hollow porous structure, which can effectively improve the energy loss of EM wave in the process of broadcasting. has been revised as The hysteresis loss and magnetic resonance of bimagnetic metals lead to the magnetic loss of the composites, and the doping of Fe further increases the hysteresis loss. On the other hand, the absorbed EM wave is reflected and scattered in the hollow porous structure, which greatly enhances the interface polarization and dielectric loss of the material.

(Page 15. Line 492): Added “The pyrolysis temperature mainly affects the graphitization degree of carbon matrix and the microstructure of MAMs. The graphitization degree is positively correlated with pyrolysis temperature; higher graphitization degree can provide greater electrical conductivity, which is conducive to the conductivity loss of incident microwave. In the process of pyrolysis, a large number of oxygen vacancies and defects at the boundary of the lower graphitized carbon layer are also conducive to the enhancement of impedance matching and polarization relaxation.

(Page 16. Line 498): With the increase of sintering temperature, organic matter is lost and the surface of magnetic nanoparticles is concave. At the same time, with the increase of particle size, the gap between particles becomes smaller and agglomeration is more likely to occur (as shown in Fig. 12(a1)). has been revised as When the microwave is incident on the surface of MAMs, the induced currents are stimulated, leading to enhanced reflection, and weaker absorption, resulting in worse microwave absorption performance, which is called "skin effect". On the other hand, surplus conductivity can interact with the incident microwave, to form an induced magnetic field in the high-frequency magnetic field, resulting in magnetic energy radiation and negative magnetic loss. In addition, the ultra-high pyrolysis temperature might lead to the loss of organic matter, the surface depression of magnetic nanoparticles, the collapse of MOF skeletons, and the smaller gap between particles, which are more prone to agglomeration, which is not conducive to multiple reflection and electromagnetic synergy (as shown in Fig. 12 (a1)).

(Page 16. Line 509): Added “According to the waveguide theory, when the wavelength is twice the length of the waveguide cross section, the microwave can be dissipated in the waveguide. Therefore, different pore sizes correspond to different microwave length, and the gradient distribution of pore structure is also conducive to broaden the EAB.

  1. Factors that govern the MOF’s selection and performance should be discussed.

A: Thank you very much for your suggestion. The choice of MOF precursor is one of the important factors that determine the performance of MAMs. After consideration, we added some new contents and introduced the characteristics of different MOFs. The specific amendments are as follows:

 (Page 9. Line 271-272): Single-metal MOFs composite has a relatively simple structure, making it convenient for the initial studies. has been revised as Cobalt-based MOF precursors have been widely used. The microwave absorbing performance of MAM is highly dependent on the morphology of MOF precursors. As a typical cobalt-based MOF, ZIF-67 has high application potential in microwave absorbing field due to its ultra-high porosity. However, the morphology of cobalt-based MOF is not limited to diamond dodecahedron.

(Page 9. Line 284): Added “The microwave absorption properties of Nickel-based MOF derivatives derived from the synergistic effect of multiple reflection, magnetic loss, conductive loss and dipole polarization.

(Page 9. Line 290): Added “Zr-based MOF derivatives have excellent chemical stability and environmental friendliness. Due to their low conductivity, they can neutralize the excessive conductivity loss of graphitized carbon, so they play a key role in achieving good impedance matching.

(Page 10. Line 318): Added “The absorption properties of Zn-based MOF derived MAMs are limited by the absence of magnetism. Therefore, few studies have been focused on pure Zn-contained MOFs derived MAMs. However, the introduction of magnetic components into Zn-containing MOFs and the combination of Zn and magnetic metals as hybrid centers to form multi-metal MOFs as MAM precursors have attracted considerable research interest in the field of microwave absorption.

  1. The literature should be summarized in several tables to be easy for the readers.

A: Two charts in the manuscript summarize some of the literature, The details are as follows. The reference was listed in the table. The information in some literature was also summarized in Figure 15, and the serial number of the corresponding literature was marked in the figure

(Page 7-8. Line 241):

Table 2. Summarizes the absorptive properties of absorbent materials (include sample a to h)

Absorbing Agent

RLmin (dB)

Maximum Peak Position (GHz)

Frequency Range (GHz) (RL ≤ -10 dB)

Efficient Bandwidth (GHz)

Effective absorption frequency(<8 GHz)

EAB range

(5 GHz)

Characteristic

Refs

(a)

ZrO2/C-800

-58.7

16.8

11.5–17.0

5.5

plentiful, dense, and equally distributed

[65]

(b)

Fe3O4@NPC

-65.5

-50

9.8

6.9

7.7–12.2

5.4–8.2

4.5

2.8

porous

[66]

(c)

Ni@C-ZIF

-86.8

13.2

6.1–8.1

2

spherical-like hierarchical 3D nanostructures

[67]

(d)

CoNi/C-650

-74.7

15.6

10.9–13.2

3.3

multi-metal

[68]

(e)

FeCoNi-MOF-600

FeCoNi-MOF-700

-23.4

-69.3

6.0

5.52

5.1–7.9

4.7–7.2

2.8

2.5

multi-metal

[69]

(f)

Porous CZC-800

−21.60

5.5

4.9–7.0

2.1

porous

[70]

(g)

CNC-1:1

-44.8

10.7

6–7.8

1.8

porous and hollow

[71]

(h)

Co3O4/Co/RGO

-52.8

13.12

-

-

two-dimensional growth

[72]

(i)

Co/C-RGO

-52

-27

9.6

13.48

7.2-13

10.3-18

5.8

7.7

two-dimensional growth

[73]

(Page 19. Line 559-560):

Figure 15. Comparison of EM wave absorption properties of MOF absorbing materials.

  1. Revise Figure 6, benzoic acid, is not the linker for PCN-222.

A: According to the literature (Feng, D.; Gu, Z. Y.; Li, J. R.; Jiang, H. L.; Wei, Z.; Zhou, H. C. Zirconiummetalloporphyrin PCN222: mesoporous metalorganic frameworks with ultrahigh stability as biomimetic catalysts. Angew. Chem. Int. Ed. 2012, 51, 10307-10310. ), the linker for PCN-222 should be Fe-TCPP. The corresponding modification chart is as follows:

(Page 8. Line 256-259):

Figure 6. Structures of main types of MOFs. (a) MIL-100; (b) HKUST-1/MOF-199; (c) MOF-74; Copyright 2011, Elsevier. (d) IRMOF-1/MOF-5; (e) PCN-222; Copyright 2012, John Wiley and Sons. (f) UIO-66; (g) ZIF-8/67.

  1. The figures’ captions should be revised. The permission for the figure reprinted from the literature should be obtained and added to the caption.

A: The referenced pictures have been applied for and licensed, and copyright has been added to the caption (Figure 1−3 is drawn by ourselves, so there is no relevant permission), the details are as follows. 37 copyright files were verified again to ensure that the file name corresponds to the picture.

(Page 6. Line 213-218): Figure 4. Schematic of physical and chemical composition of materials. Image for “ball-milling”: Copyright 2017, Elsevier. Image for “ultrasound”: Copyright 2021, Elsevier. mage for “ionic liquids”: Copyright 2020, MDPI. Image for “MIL-88@ZIF-67”: Reproduced under the terms of the CC-BY Creative Commons Attribution 4.0 International license (https://linkspringer.53yu.com/article/10.1007/s40820-018-0197-1) [57]. Image for “Au–Ni–ZnO hybrid nanocrystals”: Copyright 2015, Small.

(Page 7. Line 239-240): Figure 5 The EAB is broadened by the combination of a variety of MOF derivatives. (c) Copyright 2019, American Chemical Society. (g) Copyright 2019, Elsevier. (i) Copyright 2017, Elsevier.

(Page 8. Line 257-259): Figure 6. Structures of main types of MOFs. (a) MIL-100; (b) HKUST-1/MOF-199; (c) MOF-74; Copyright 2011, Elsevier. (d) IRMOF-1/MOF-5; (e) PCN-222; Copyright 2012, John Wiley and Sons. (f) UIO-66; (g) ZIF-8/67.

(Page 9. Line 291-297): Figure 7. Preparation and morphology of single-metal MOFs microwave absorbing materials. (a) The synthesis process of CNC nanoparticles. Copyright 2022, Elsevier. (b) the synthesis of earthworm-like (Co/CoO)@C composite. Copyright 2021, Elsevier. (c) the synthesis of Ni-MOF hollow spheres with controllable surface architecture. Copyright 2019, ACS Publications. (d) diagram and SEM of UIO-66. SEM on the left: Copyright 2014, Elsevier. SEM on the right: Reproduced under the terms of the CC-BY Creative Commons Attribution 4.0 International license (https://linkspringer.53yu.com/article/10.1007/s40820-021-00606-6) [94].

(Page 11. Line 330-334): Figure 8. (a) Schematic diagram of the EM wave absorption mechanism for CoNi/C nanocomposites.  (b) porous CZC composites. Copyright 2018, ACS Publications. (c) porous flower-like Zinc-doped Ni-MOF precursor. Copyright 2019, Elsevier. (d) the synthetic process of hollow CNC microspheres and SEM images of CoNi-MOFs and CNC-1:1. Copyright 2019, Elsevier. (e) the synthesis process and formation mechanism of Co@ZnO@NC-a, Co@ZnO@NC-b, Co@NC. Copyright 2022, Elsevier.

(Page 13. Line 391-397): Figure 9. (a) The synthesis strategy of FMCFs composite. Copyright 2021, Elsevier. (b) the synthesis process of CoNi@NCPs-rGO. Copyright 2021, Elsevier. and the SEM images of MOF/GO hybrid. Copyright 2017, Elsevier. (c) the formation process of Co–Fe alloy/N-doped carbon hollow spheres and FESEM images of Co–Fe/NC-700 hollow spheres. Copyright 2019, Wiley Online Library. (d) the synthesis process of NiFe2S4/PC. Copyright 2022, Elsevier. (e) the synthesis process of ZnO/C@PG. Copyright 2021, Elsevier. (f) the synthesis of CF@ZIF-67 and CF@C/Co(carbonized at 700 °C) and their SEM images. Copyright 2021, Elsevier.

(Page 14. Line 418-421): Figure 10 (a1) The synthetic process of UM-700 and DM-700, and the XRD patterns of DM- 700. Copyright 2021, Elsevier. (a2) the synthetic process of DMM-700 and the SEM image of DM. Copyright 2022, Elsevier. (b) the synthetic process of Cu/NC@Co/NC composites. Copyright 2022, Elsevier. (c) synthetic process of Co/MOS/NPC composites. Copyright 2021, Elsevier.

(Page 15. Line 454-459): Figure 11 (a) The synthesis process of CFMM. Copyright 2022, Elsevier. (b) the synthesis of Z67-8@C and TEM image of Z67-8@C. Copyright 2021, Elsevier. (c1) the synthesis process of hollow ZIF. Copyright 2016, John Wiley and Sons. (c2) and ZIF-67@ZIF8/Ni2+. Copyright 2021, Elsevier. (d) the synthetic process of yolk-shell Ni@C@ZnO microspheres. Copyright 2020, Elsevier. (e) the synthesis procedure of the FeCoNi@C nanocomposites and the SEM image of FeCoNi-MOF-900, RL of FeCoNi-MOF-700. Copyright 2019, American Chemical Society.

(Page 16. Line 493-496): Figure 12 (a) TG curves of the Co-MOF (ZIF-67) precursors under air and N2 and (a1) SEM images of Co/C-500, Co/C-600 and Co/C-800. Copyright 2015, American Chemical Society Publications (b) the formation of PCHMs and SEM images of PCHMs (carbonized at 500 °C, 550 °C and 750 °C). Copyright 2019, Elsevier.

(Page 17. Line 521-524): Figure 13 (a) S-Co/C-0.1, S-Co/C-0.3 and S-Co/C-0.5. Copyright 2018, American Chemical Society; (b) Schematic preparation process of Ni@NC com posites and the SEM images of the precursors with different dosage of pyridine. Copyright 2022, Elsevier. (c) SEM images and elemental mappings of CFZC-0.1. Copyright 2021, Elsevier.

(Page 18. Line 545-550): Figure 14 (a) The synthesis process of FeMZn-MOFs and SEM images of FeCoZn MOFs: Fe:Co = 6:1 (a1-a2), Fe:Co = 4:1 (a3-a4) and Fe Co = 2:3 (a5-a6); FeNiZn MOFs: Fe:Ni = 6:1 (a7-a8), Fe: Ni = 4:1 (a9-a10) and Fe: Ni = 2:3 (a11-a12). Copyright 2022, Elsevier. (b) the synthesis process of Co@NC composites and SEM images of Co@NC-f with ZIF-67-f as the precursor (b1), Co@NCNT-d with ZIF-67-d as the precursor (b2), Co@NC-s with ZIF-67-s as the precursor(b3), Co@NCNT-o with ZIF-67-o as the precursor (b4), Co@NCNT-c with ZIF-67-c as the precursor(b5). Copyright 2020, Small.

  1. References for MOFs should be updated, including these Reviews;

 https://doi.org/10.1016/j.ccr.2021.214263;

 https://doi.org/10.1016/j.chemosphere.2022.134184; 

https://doi.org/10.1016/j.surfin.2021.101647; 

DOI: 10.1039/D1TA06006F

A: In the “6. Challenges and Prospects”, 4 recently published reviews on MOF are added, and the corresponding content has been modified as follows:

(Page 19. Line 584-587) has been revised as:

In addition, the application of MOF is not limited to the field of microwave absorption, MOFs@MOFs, MOFs of core-shell structure and multilayer core-shell structure, CelloMOFs [128], Zr-MOF [129], MOF/COF-based hybrid nanomaterials [130] and other MOF materials are also widely used in water treatment, air purification, biology, energy storage, catalysts, membrane separation [128-131], supercapacitors [128, 132] and other fields, which have broad development prospects.

(Page 25. Line 900), the following 4 references have been added

[128] Abdelhamid, H. N.; Mathew, A. P. Cellulose–metal organic frameworks (CelloMOFs) hybrid materials and their multifaceted Applications: A review. Coord. Chem. Rev. 2022, 451, 214263.

[129] Du, Q.; Rao, R.; Bi, F.; Yang, Y.; Zhang, W.; Yang, Y.; Zhang, W.; Yang, Y.; Liu, N.; Zhang, X. Preparation of modified zirconium-based metal-organic frameworks (Zr-MOFs) supported metals and recent application in environment: A review and perspectives. Surf. Interfaces, 2022, 28, 101647.

[130] Guo, C.; Duan, F.; Zhang, S.; He, L.; Wang, M.; Chen, J.; Zhang, J.; Jia, Q.; Zhang, Z.; Du, M. Heterostructured hybrids of metal–organic frameworks (MOFs) and covalent–organic frameworks (COFs). J. Mater. Chem. A, 2022, 10, 475-507.

[131] Annamalai, J.; Murugan, P.; Ganapathy, D.; Nallaswamy, D.; Atchudan, R.; Arya, S.; Khosla, A.; Barathi, S.; Sundramoorthy, A. K. Synthesis of various dimensional metal organic frameworks (MOFs) and their hybrid composites for emerging applications–a review. Chemosphere, 2022, 134184.

  1. The language should be revised, and typos should be corrected.

A: We have re-checked the manuscript and corrected several grammatical errors.

(Page 2. Line 48): the bandwidth of absorbing frequencies impedance matching regulation has been revised as the bandwidth of absorbing frequencies is impedance matching regulation.

(Page 4. Line 137): which due to has been revised as which is due to.

(Page 4. Line 139): which mainly absorb has been revised as which are mainly absorbed.

(Page 4. Line 148): this materials has been revised as these materials.

(Page 6. Line 214): are combined together has been revised as are combined.

(Page 9. Line 280): performance of as-prepared composite has been revised as performance of the as-prepared composite.

(Page 9. Line 291): Benefiting from has been revised as Benefit from.

(Page 11. Line 353): MOF derivative has been revised as MOF derivatives.

(Page 12. Line 366): conductive current has been revised as a conductive current.

(Page 12. Line 381): the ZIF-67 nanoparticles structure has been revised as the structure of ZIF-67 nanoparticles .

(Page 14. Line 447): demonstrate has been revised as demonstrated.

(Page 16. Line 517): different core-shell thickness, different pore size, different interface structure has been revised as different core-shell thicknesses, different pore sizes, different interface structures.

  1. Adjust dash for ‘1.2~8 GHz’; ‘1MHz~50MHz’; ‘10~25 nm’;

A: 17 dashes have been Adjusted based on published articles on Coatings.

(Page 1. Line 45): 8~12 GHz or 8~18 GHz has been revised as 8−12 GHz or 8−18 GHz.

(Page 2. Line 47): 1.2~8 GHz has been revised as 1.2−8 GHz.

(Page 4. Line 137): 0.1 GHz~1 THz has been revised as 0.1 GHz−1 THz.

(Page 4. Line 143): 1MHz~50MHz has been revised as 1 MHz−50 MHz.

(Page 5. Line 189 and Line 190): 15~25 nm has been revised as 15−25 nm.

(Page 5. Line 194): 1~8 GHz is about 10~25 nm has been revised as 1−8 GHz is about 10−25 nm.

(Page 9. Line 268): 4.8~6.2 GHz has been revised as 4.8−6.2 GHz.

(Page 9. Line 274): 0.43~1.88 GHz has been revised as 0.43−1.88 GHz.

(Page 9. Line 284): 3.4~18.0 GHz has been revised as 3.4−18.0 GHz.

(Page 9. Line 285): 1.0~5.0 mm has been revised as 1.0−5.0 mm.

(Page 9. Line 287): 4~11.4 GHz has been revised as 4−11.4 GHz.

(Page 16. Line 520): 4~7.8 GHz has been revised as 4−7.8 GHz.

(Page 17. Line 543): 5.5~7.75 GHz has been revised as 5.5−7.75 GHz.

(Page 17. Line 544): 6.75~4.75 GHz has been revised as 6.75−4.75 GHz.

  1. References and copyright for Figure 2 and Figure 3 should be added.

A: All the content on Figures 2 and 3 has been prepared and drawn by learning from key points in the literature and combining them with our own understanding, so the references and copyright have not been added.

  1. Change writing style of ‘Zhang's research team’; ‘Rcsle's research team’; ‘Li's research team’, ‘Cui's research team’, ….. to ‘last name et al.’ or ’coworks’.

A: 13 similar issues in the manuscript have been revised as follows.

(Page 9. Line 264): Liu's research team has been revised as Liu et al..

(Page 9. Line 275): Yang's research team has been revised as Yang et al..

(Page 10. Line 299): Liu's research team has been revised as Liu et al..

(Page 10. Line 318): Liu's research team has been revised as Liu et al..

(Page 10. Line 323): Yang's research team has been revised as Yang et al..

(Page 10. Line 344): Chen's research team has been revised as Chen et al..

(Page 12. Line 357): Zhang's research team has been revised as Zhang et al..

(Page 12. Line 364): Zhang's research team has been revised as Zhang et al..

(Page 12. Line 366): Yuan's research team has been revised as Yuan et al..

(Page 12. Line 371): Zhang's research team has been revised as Zhang et al..

(Page 12. Line 377): Song's research team has been revised as Song et al..

(Page 12. Line 385-386): Tao's research team has been revised as Tao et al..

(Page 13. Line 400): Wu's research team has been revised as Wu et al..

(Page 13. Line 413): Peng's research team has been revised as Peng et al..

(Page 14. Line 432): Li's research team has been revised as Li et al..

(Page 14. Line 433): Rçsle's research team has been revised as Rösler et al..

(Page 14. Line 439): Cui's research team has been revised as Cui et al..

(Page 14. Line 443): Wang's research team has been revised as Wang et al..

(Page 15. Line 476): Liu's research team has been revised as Liu et al..

(Page 16. Line 503): Liu's research team has been revised as Liu et al..

(Page 16. Line 508): Jin's research team has been revised as Jin et al..

(Page 16. Line 517): Kong's research team has been revised as Kong et al..

(Page 17. Line 526): Wu's research team has been revised as Wu et al..

  1. Revise typos such as ‘Co-MOF derived (CO/CoO)@C was’; 

(Page 9. Line 269): Co-MOF derived (CO/CoO)@C was has been revised as Co-MOF derived (CO/CoO)@C composite was.

In addition to the modifications to the above suggestions, we carefully checked the correctness of the references and made the following modifications.

  1. (Page 1. Line 27): Changed the font of “[1-10]” to Palatino Linotype.
  2. (Page 1. Line 30): Changed the font of “[11-12]” to Palatino Linotype.
  3. (Page 7. Line 241): “RLMin” has been revised as “RLmin”.
  4. (Page 9. Line 276): “600” has been revised as “600 °C”.
  5. Cut (Page 9. Line 284-298) to (Page 9. Line 290).
  6. Cut (Page 12. Line 375-382) to (Page 12. Line 372).
  7. (Page 12. Line 376): “Figure 9 (c)” has been revised as “Figure 9 (b)”.
  8. (Page 13. Line 397): “700” has been revised as “700 °C”.
  9. (Page 15. Line 455): Deleted the two spaces before “Figure 11”.
  10. (Page 15. Line 456): “Wiley Online Library” has been revised as “John Wiley and Sons”.
  11. (Page 16. Line 484): “500” has been revised as “500 °C”.
  12. (Page 24. Line 861-862): [113] were cited incorrectly. “[113] Cui, Y.; Liu, Z.; Li, X.; Ren, J.; Wang, Y.; Zhang, Q.; Zhang, B. MOF-derived yolk-shell Co@ZnO/Ni@NC nanocage: Structure control and electromagnetic wave absorption performance. J. Colloid. Interf. Sci. 2021, 600, 99-110.” has been revised as “[113] Li, S.; Lin, L.; Yao, L.; Zheng, H.; Luo, Q.; Xu, W.; Zhang, C.; Xie, Q.; Wang, L. S.; Peng, D. L. MOFs-derived Co-C@ C hollow composites with high-performance electromagnetic wave absorption. J. Alloy. Compd. 2021, 856, 158183.
  13. (Page 25. Line 892-893): “[128]” has been revised as “[132]”.
  14. Two copyright permission documents in Figure 6 have been applied, but forgot to upload. We have been added to the new uploaded file.

We really hope that the revisions and responses are made the paper better now, and we will not hesitate to revise the paper if there are any farther comments later.

Thanks again for the reviewer’s precious comments and careful corrections.

Sincerely yours,

Qingwei Liao

Reviewer 2 Report

L. Qin and co-workers revised and enhanced considerably their proposed review, now entitled ‘Low Frequency Broadband Absorbing Coatings Based on MOFs: Design, Fabrication, Microstructure and Properties’ as I suggested. Indeed, almost all my suggestions were attended in satisfactory manner. For instance, all figures have now enough sharpness, a new conclusion remarks section apart from Challenges and Prospects section was included, and as I mentioned before, the tittle was modified as I suggested. For these reasons, I recommend publication of this manuscript in its present form.

Author Response

(The authors gave the same response as above.)

Reviewer 3 Report

The article contains 128 literary references, which testifies to the great work of the authors on the analysis of data accumulated in the world on the creation and use of materials that absorb microwaves.

Given the quality and nature of the data presented, I believe that this article will be of interest to a wide audience in this field and can be published without significant changes.

Author Response

(The authors gave the same response as above.)

Reviewer 4 Report

 The authors reviewed applications of MOFs as adsorbents for low-Frequency Broadband. The submission is accepted for publication after considering the following points:-

1.      An in-depth discussion regarding the topic should be included.

2.      Factors that govern the MOF’s selection and performance should be discussed.

3.      The literature should be summarized in several tables to be easy for the readers.

4.      Revise Figure 6, benzoic acid, is not the linker for PCN-222.

5.      The figures’ captions should be revised. The permission for the figure reprinted from the literature should be obtained and added to the caption.

6.      References for MOFs should be updated, including these Reviews; https://doi.org/10.1016/j.ccr.2021.214263; https://doi.org/10.1016/j.chemosphere.2022.134184; https://doi.org/10.1016/j.surfin.2021.101647; DOI: 10.1039/D1TA06006F

7.      The language should be revised, and typos should be corrected.

Minors

8.       Adjust dash for ‘1.2~8 GHz’; ‘1MHz~50MHz’; ‘10~25 nm’;

9.       References and copyright for Figure 2 and Figure 3 should be added.

10.  Change writing style of ‘Zhang's research team’; ‘Rcsle's research team’; ‘Li's research team’, ‘Cui's research team’, ….. to ‘last name et al.’ or ’coworks’

11.  Revise typos such as ‘Co-MOF derived (CO/CoO)@C was’; 

Author Response

(The authors gave the same response as above.)
